# CLASSIFICATION FROM POSITIVE, UNLABELED AND BIASED NEGATIVE DATA

## ABSTRACT

Positive-unlabeled (PU) learning addresses the problem of learning a binary classifier from positive (P) and unlabeled (U) data. It is often applied to situations where negative (N) data are difficult to be fully labeled. However, collecting a non-representative N set that contains only a small portion of all possible N data can be much easier in many practical situations. This paper studies a novel classification framework which incorporates such biased N (bN) data in PU learning. The fact that the training N data are biased also makes our work very different from those of standard semi-supervised learning. We provide an empirical risk minimization-based method to address this PUbN classification problem. Our approach can be regarded as a variant of traditional example-reweighting algorithms, with the weight of each example computed through a preliminary step that draws inspiration from PU learning. We also derive an estimation error bound for the proposed method. Experimental results demonstrate the effectiveness of our algorithm in not only PUbN learning scenarios but also ordinary PU leaning scenarios on several benchmark datasets.

## 1 INTRODUCTION

In conventional binary classification, examples are labeled as either positive (P) or negative (N), and we train a classifier on these labeled examples. On the contrary, positive-unlabeled (PU) learning addresses the problem of learning a classifier from P and unlabeled (U) data, without need of explicitly identifying N data (Elkan & Noto, 2008; Ward et al., 2009).

PU learning finds its usefulness in many real-world problems. For example, in one-class remote sensing classification (Li et al., 2011), we seek to extract a specific land-cover class from an image. While it is easy to label examples of this specific land-cover class of interest, examples not belonging to this class are too diverse to be exhaustively annotated. The same problem arises in text classification, as it is difficult or even impossible to compile a set of N samples that provides a comprehensive characterization of everything that is not in the P class (Liu et al., 2003; Fung et al., 2006). Besides, PU learning has also been applied to other domains such as outlier detection (Hido et al., 2008; Scott & Blanchard, 2009), medical diagnosis (Zuluaga et al., 2011), or time series classification (Nguyen et al., 2011).

By carefully examining the above examples, we find out that the most difficult step is often to collect a fully representative N set, whereas only labeling a small portion of all possible N data is relatively easy. Therefore, in this paper, we propose to study the problem of learning from P, U and biased N (bN) data, which we name PUbN learning hereinafter. We suppose that in addition to P and U data, we also gather a set of bN samples, governed by a distribution distinct from the true N distribution. As described previously, this can be viewed as an extension of PU learning, but such bias may also occur naturally in some real-world scenarios. For instance, let us presume that we would like to judge whether a subject is affected by a particular disease based on the result of a physical examination. While the data collected from the patients represent rather well the P distribution, healthy subjects that request the examination are in general highly biased with respect to the whole healthy subject population.

We are not the first to be interested in learning with bN data. In fact, both Li et al. (2010) and Fei & Liu (2015) attempted to solve similar problems in the context of text classification. Li et al. (2010) simply discarded negative samples and performed ordinary PU classification. It was also

mentioned in the paper that bN data could be harmful. Fei & Liu (2015) adapted another strategy. The authors considered even gathering unbiased U data is difficult and learned the classifier from only P and bN data. However, their method is specific to text classification because it relies on the use of effective similarity measures to evaluate similarity between documents. Therefore, our work differs from these two in that the classifier is trained simultaneously on P, U and bN data, without resorting to domain-specific knowledge. The presence of U data allows us to address the problem from a statistical viewpoint, and thus the proposed method can be applied to any PUbN learning problem in principle.

In this paper, we develop an empirical risk minimization-based algorithm that combines both PU learning and importance weighting to solve the PUbN classification problem, We first estimate the probability that an example is sampled into the P or the bN set. Based on this estimate, we regard bN and U data as N examples with instance-dependent weights. In particular, we assign larger weights to U examples that we believe to appear less often in the P and bN sets. P data are treated as P examples with unity weight but also as N examples with usually small or zero weight whose actual value depends on the same estimate.

The contributions of the paper are three-fold:

1. We formulate the PUbN learning problem as an extension of PU learning and propose an empirical risk minimization-based method to address the problem. We also theoretically establish an estimation error bound for the proposed method.

2. We experimentally demonstrate that the classification performance can be effectively improved thanks to the use of bN data during training. In other words, PUbN learning yields better performance than PU learning.

3. Our method can be easily adapted to ordinary PU learning. Experimentally we show that the resulting algorithm allows us to obtain new state-of-the-art results on several PU learning tasks.

**Relation with Semi-supervised Learning** With P, N and U data available for training, our problem setup may seem similar to that of semi-supervised learning (Chapelle et al., 2010; Oliver et al., 2018). Nonetheless, in our case, N data are biased and often represent only a small portion of the whole N distribution. Therefore, most of the existing methods designed for the latter cannot be directly applied to the PUbN classification problem. Furthermore, our focus is on deducing a risk estimator using the three sets of data, whereas in semi-supervised learning the main concern is often how U data can be utilized for regularization (Grandvalet & Bengio, 2005; Belkin et al., 2006; Laine & Aila, 2017; Miyato et al., 2016). The two should be compatible and we believe adding such regularization to our algorithm can be beneficial in many cases.

**Relation with Dataset Shift** PUbN learning can also be viewed as a special case of dataset shift[1] (Quionero-Candela et al., 2009) if we consider that P and bN data are drawn from the training distribution while U data are drawn from the test distribution. Covariate shift (Shimodaira, 2000; Sugiyama & Kawanabe, 2012) is another special case of dataset shift that has been studied intensively. In the covariate shift problem setting, training and test distributions have the same class conditional distribution and only differ in the marginal distribution of the independent variable. One popular approach to tackle this problem is to reweight each training example according to the ratio of the test density to the training density (Huang et al., 2007; Sugiyama et al., 2008). Nevertheless, simply training a classifier on a reweighted version of the labeled set is not sufficient in our case since there may be examples with zero probability to be labeled. It is also important to notice that the problem of PUbN learning is intrinsically different from that of covariate shift and neither of the two is a special case of the other.

## 2 PROBLEM SETTING

In this section, we briefly review the formulations of PN, PU and PNU classification and introduce the problem of learning from P, U and bN data.

---

[1] Dataset shift refers to any case where training and test distributions differ. The term sample selection bias (Heckman, 1979; Zadrozny, 2004) is sometimes used to describe the same thing. However, strictly speaking, sample selection bias actually refers to the case where training instances are first drawn from the test distributions and then a subset of these data is systematically discarded due to a particular mechanism.

## 2.1 Standard Binary Classification

Let $\boldsymbol{x} \in \mathbb{R}^d$ and $y \in \{+1, -1\}$ be random variables following an unknown probability distribution with density $p(\boldsymbol{x}, y)$. Let $g : \mathbb{R}^d \to \mathbb{R}$ be an arbitrary decision function for binary classification and $\ell : \mathbb{R} \to \mathbb{R}_+$ be a loss function of margin $yg(\boldsymbol{x})$ that usually takes a small value for a large margin. The goal of binary classification is to find $g$ that minimizes the classification risk:

$$R(g) = \mathbb{E}_{(\boldsymbol{x},y) \sim p(\boldsymbol{x},y)}[\ell(yg(\boldsymbol{x}))], \tag{1}$$

where $\mathbb{E}_{(\boldsymbol{x},y) \sim p(\boldsymbol{x},y)}[\cdot]$ denotes the expectation over the joint distribution $p(\boldsymbol{x}, y)$. When we care about classification accuracy, $\ell$ is the zero-one loss $\ell_{01}(z) = (1 - \text{sign}(z))/2$. However, for ease of optimization, $\ell_{01}$ is often substituted with a surrogate loss such as the sigmoid loss $\ell_{\text{sig}}(z) = 1/(1 + \exp(z))$ or the logistic loss $\ell_{\log}(z) = \ln(1 + \exp(-z))$ during learning.

In standard supervised learning scenarios (PN classification), we are given P and N data that are sampled independently from $p(\boldsymbol{x} \mid y = +1)$ and $p(\boldsymbol{x} \mid y = -1)$ as $\mathcal{X}_{\text{P}} = \{\boldsymbol{x}_i^{\text{P}}\}_{i=1}^{n_{\text{P}}}$ and $\mathcal{X}_{\text{N}} = \{\boldsymbol{x}_i^{\text{N}}\}_{i=1}^{n_{\text{N}}}$. Let us denote by $R_{\text{P}}^+(g) = \mathbb{E}_{\boldsymbol{x} \sim p(\boldsymbol{x}|y=+1)}[\ell(g(\boldsymbol{x}))]$, $R_{\text{N}}^-(g) = \mathbb{E}_{\boldsymbol{x} \sim p(\boldsymbol{x}|y=-1)}[\ell(-g(\boldsymbol{x}))]$ partial risks and $\pi = p(y = 1)$ the P prior. We have the equality $R(g) = \pi R_{\text{P}}^+(g) + (1 - \pi)R_{\text{N}}^-(g)$. The classification risk (1) can then be empirically approximated from data by

$$\hat{R}_{\text{PN}}(g) = \pi \hat{R}_{\text{P}}^+(g) + (1 - \pi)\hat{R}_{\text{N}}^-(g),$$

where $\hat{R}_{\text{P}}^+(g) = \frac{1}{n_{\text{P}}}\sum_{i=1}^{n_{\text{P}}} \ell(g(\boldsymbol{x}_i^{\text{P}}))$ and $\hat{R}_{\text{N}}^-(g) = \frac{1}{n_{\text{N}}}\sum_{i=1}^{n_{\text{N}}} \ell(-g(\boldsymbol{x}_i^{\text{N}}))$. By minimizing $\hat{R}_{\text{PN}}(g)$ we obtain the ordinary empirical risk minimizer $\hat{g}_{\text{PN}}$.

## 2.2 PU Classification

In PU classification, instead of N data $\mathcal{X}_{\text{N}}$ we have only access to $\mathcal{X}_{\text{U}} = \{x_i^{\text{U}}\}_{i=1}^{n_{\text{U}}} \sim p(\boldsymbol{x})$ a set of U samples drawn from the marginal density $p(\boldsymbol{x})$. Several effective algorithms have been designed to address this problem. Liu et al. (2002) proposed the S-EM approach that first identifies reliable N data in the U set and then runs the Expectation-Maximization (EM) algorithm to build the final classifier. The biased support vector machine (Biased SVM) introduced in Liu et al. (2003) regards U samples as N samples with smaller weights. Mordelet & Vert (2014) solved the PU problem by aggregating classifiers trained to discriminate P data from a small random subsample of U data.

More recently, attention has been paid on the unbiased risk estimator proposed in du Plessis et al. (2014) and du Plessis et al. (2015). The key idea is to use the following equality:

$$(1 - \pi)R_{\text{N}}^-(g) = R_{\text{U}}^-(g) - \pi R_{\text{P}}^-(g),$$

where $R_{\text{U}}^-(g) = \mathbb{E}_{x \sim p(\boldsymbol{x})}[\ell(-g(\boldsymbol{x}))]$ and $R_{\text{P}}^-(g) = \mathbb{E}_{x \sim p(\boldsymbol{x}|y=+1)}[\ell(-g(\boldsymbol{x}))]$. This equality is acquired by exploiting the fact $p(\boldsymbol{x}) = \pi p(\boldsymbol{x} \mid y = +1) + (1 - \pi)p(\boldsymbol{x} \mid y = -1)$. As a result, we can approximate the classification risk (1) by

$$\hat{R}_{\text{PU}}(g) = \pi \hat{R}_{\text{P}}^+(g) - \pi \hat{R}_{\text{P}}^-(g) + \hat{R}_{\text{U}}^-(g), \tag{2}$$

where $\hat{R}_{\text{P}}^-(g) = \frac{1}{n_{\text{P}}}\sum_{i=1}^{n_{\text{P}}} \ell(-g(\boldsymbol{x}_i^{\text{P}}))$ and $\hat{R}_{\text{U}}^-(g) = \frac{1}{n_{\text{U}}}\sum_{i=1}^{n_{\text{U}}} \ell(-g(\boldsymbol{x}_i^{\text{U}}))$. We then minimize $\hat{R}_{\text{PU}}(g)$ to obtain another empirical risk minimizer $\hat{g}_{\text{PU}}$. Note that as the loss is always positive, the classification risk (1) that $\hat{R}_{\text{PU}}(g)$ approximates is also positive. However, Kiryo et al. (2017) pointed out that when the model of $g$ is too flexible, that is, when the function class $\mathcal{G}$ is too large, $\hat{R}_{\text{PU}}(\hat{g}_{\text{PU}})$ indeed goes negative and the model seriously overfits the training data. To alleviate overfitting, the authors observed that $R_{\text{U}}^-(g) - \pi R_{\text{P}}^-(g) = (1 - \pi)R_{\text{N}}^-(g) \geq 0$ and proposed the non-negative risk estimator for PU learning:

$$\tilde{R}_{\text{PU}}(g) = \pi \hat{R}_{\text{P}}^+(g) + \max\{0, \hat{R}_{\text{U}}^-(g) - \pi \hat{R}_{\text{P}}^-(g)\}. \tag{3}$$

In terms of implementation, stochastic optimization was used and when $r = \hat{R}_{\text{U}}^-(g) - \pi \hat{R}_{\text{P}}^-(g)$ becomes negative for a mini-batch, they performed a step of gradient ascent along $\nabla r$ to make the mini-batch less overfitted.

## 2.3 PNU CLASSIFICATION

In semi-supervised learning (PNU classification), P, N and U data are all available. An abundance of works have been dedicated to solving this problem. Here we in particular introduce the PNU risk estimator proposed in Sakai et al. (2017). By directly leveraging U data for risk estimation, it is the most comparable to our method. The PNU risk is simply defined as a linear combination of PN and PU/NU risks. Let us just consider the case where PN and PU risks are combined, then for some $\gamma \in [0, 1]$, the PNU risk estimator is expressed as

$$
\begin{aligned}
\hat{R}^{\gamma}_{\text{PNU}}(g) &= \gamma \hat{R}_{\text{PN}}(g) + (1 - \gamma)\hat{R}_{\text{PU}}(g) \\
&= \pi \hat{R}^+_{\text{P}}(g) + \gamma(1 - \pi)\hat{R}^-_{\text{N}}(g) + (1 - \gamma)(\hat{R}^-_{\text{U}}(g) - \pi \hat{R}^-_{\text{P}}(g)).
\end{aligned} \tag{4}
$$

We can again consider the non-negative correction by forcing the term $\gamma(1 - \pi)\hat{R}^-_{\text{N}}(g) + (1 - \gamma)(\hat{R}^-_{\text{U}}(g) - \pi \hat{R}^-_{\text{P}}(g))$ to be non-negative. In the rest of the paper, we refer to the resulting algorithm as non-negative PNU (nnPNU) learning (see Appendix D.4 for an alternative definition of nnPNU and the corresponding results).

## 2.4 PUbN CLASSIFICATION

In this paper, we study the problem of PUbN learning. It differs from usual semi-supervised learning in the fact that labeled N data are not fully representative of the underlying N distribution $p(\boldsymbol{x} \mid y = -1)$. To take this point into account, we introduce a latent random variable $s$ and consider the joint distribution $p(\boldsymbol{x}, y, s)$ with constraint $p(s = +1 \mid \boldsymbol{x}, y = +1) = 1$. Equivalently, $p(y = -1 \mid \boldsymbol{x}, s = -1) = 1$. Let $\rho = p(y = -1, s = +1)$. Both $\pi$ and $\rho$ are assumed known throughout the paper. In practice they often need to be estimated from data (Jain et al., 2016; Ramaswamy et al., 2016; du Plessis et al., 2017). In place of ordinary N data we collect a set of bN samples

$$
\mathcal{X}_{\text{bN}} = \{\boldsymbol{x}^{\text{bN}}_i\}^{n_{\text{bN}}}_{i=1} \sim p(\boldsymbol{x}|y = -1, s = +1).
$$

The goal remains the same: we would like to minimize the classification risk (1).

## 3 METHOD

In this section, we propose a risk estimator for PUbN classification and establish an estimation error bound for the proposed method. Finally we show how our method can be applied to PU learning as a special case when no bN data are available.

### 3.1 RISK ESTIMATOR

Let $R^-_{\text{bN}}(g) = \mathbb{E}_{x \sim p(\boldsymbol{x}|y=-1,s=+1)}[\ell(-g(\boldsymbol{x}))]$ and $R^-_{s=-1}(g) = \mathbb{E}_{x \sim p(\boldsymbol{x}|s=-1)}[\ell(-g(\boldsymbol{x}))]$. Since $p(\boldsymbol{x}) = p(\boldsymbol{x}, y = +1) + p(\boldsymbol{x}, y = -1, s = +1) + p(\boldsymbol{x}, s = -1)$, we have

$$
R(g) = \pi R^+_{\text{P}}(g) + \rho R^-_{\text{bN}}(g) + (1 - \pi - \rho)R^-_{s=-1}(g). \tag{5}
$$

The first two terms on the right-hand side of the equation can be approximated directly from data by writing $\hat{R}^+_{\text{P}}(g) = \frac{1}{n_{\text{P}}} \sum^{n_{\text{P}}}_{i=1} \ell(g(\boldsymbol{x}^{\text{P}}_i))$ and $\hat{R}^-_{\text{bN}}(g) = \frac{1}{n_{\text{bN}}} \sum^{n_{\text{bN}}}_{i=1} \ell(-g(\boldsymbol{x}^{\text{bN}}_i))$. We therefore focus on the third term $\bar{R}^-_{s=-1}(g) := (1 - \pi - \rho)R^-_{s=-1}(g)$. Our approach is mainly based on the following theorem. We relegate all proofs to the appendix.

**Theorem 1.** *Let $\sigma(\boldsymbol{x}) = p(s = +1 \mid \boldsymbol{x})$. For all $\eta \in [0, 1]$ and $h : \mathbb{R}^d \to [0, 1]$ satisfying the condition $h(\boldsymbol{x}) > \eta \Rightarrow \sigma(\boldsymbol{x}) > 0$, the risk $\bar{R}^-_{s=-1}(g)$ can be expressed as*

$$\bar{R}_{s=-1}^{-}(g) = \mathbb{E}_{\boldsymbol{x}\sim p(\boldsymbol{x})}[\mathbb{1}_{h(\boldsymbol{x})\leq\eta}\,\ell(-g(\boldsymbol{x}))(1-\sigma(\boldsymbol{x}))]$$

$$+ \pi\,\mathbb{E}_{\boldsymbol{x}\sim p(\boldsymbol{x}|y=+1)}\left[\mathbb{1}_{h(\boldsymbol{x})>\eta}\,\ell(-g(\boldsymbol{x}))\frac{1-\sigma(\boldsymbol{x})}{\sigma(\boldsymbol{x})}\right]$$

$$+ \rho\,\mathbb{E}_{\boldsymbol{x}\sim p(\boldsymbol{x}|s=+1,y=-1)}\left[\mathbb{1}_{h(\boldsymbol{x})>\eta}\,\ell(-g(\boldsymbol{x}))\frac{1-\sigma(\boldsymbol{x})}{\sigma(\boldsymbol{x})}\right]. \tag{6}$$

In the theorem, $\bar{R}_{s=-1}^{-}(g)$ is decomposed into three terms, and when the expectation is substituted with the average over training samples, these three terms are approximated respectively using data from $\mathcal{X}_{\mathrm{U}}$, $\mathcal{X}_{\mathrm{P}}$ and $\mathcal{X}_{\mathrm{bN}}$. The choice of $h$ and $\eta$ is thus very crucial because it determines what each of the three terms tries to capture in practice. Ideally, we would like $h$ to be an approximation of $\sigma$. Then, for $\boldsymbol{x}$ such that $h(\boldsymbol{x})$ is close to 1, $\sigma(\boldsymbol{x})$ is close to 1, so the last two terms on the right-hand side of the equation can be reasonably evaluated using $\mathcal{X}_{\mathrm{P}}$ and $\mathcal{X}_{\mathrm{bN}}$ (i.e., samples drawn from $p(\boldsymbol{x}\mid s=+1)$). On the contrary, if $h(\boldsymbol{x})$ is small, $\sigma(\boldsymbol{x})$ is small and such samples can be hardly found in $\mathcal{X}_{\mathrm{P}}$ or $\mathcal{X}_{\mathrm{bN}}$. Consequently the first term appeared in the decomposition is approximated with the help of $\mathcal{X}_{\mathrm{U}}$. Finally, in the empirical risk minimization paradigm, $\eta$ becomes a hyperparameter that controls how important U data is against P and bN data when we evaluate $\bar{R}_{s=-1}^{-}(g)$. The larger $\eta$ is, the more attention we would pay to U data.

One may be curious about why we do not simply approximate the whole risk using only U samples, that is, set $\eta$ to 1. There are two main reasons. On one hand, if we have a very small U set, which means $n_{\mathrm{U}}\ll n_{\mathrm{P}}$ and $n_{\mathrm{U}}\ll n_{\mathrm{bN}}$, approximating a part of the risk with labeled samples should help us reduce the estimation error. This may seem unrealistic but sometimes unbiased U samples can also be difficult to collect (Ishida et al., 2018). On the other hand, more importantly, we have empirically observed that when the model of $g$ is highly flexible, even a sample regarded as N with small weight gets classified as N in the latter stage of training and performance of the resulting classifier can thus be severely degraded. Introducing $\eta$ alleviates this problem by avoiding treating all U data as N samples.

As $\sigma$ is not available in reality, we propose to replace $\sigma$ by its estimate $\hat{\sigma}$ in (6). We further substitute $h$ with the same estimate and obtain the following expression:

$$\bar{R}_{s=-1,\eta,\hat{\sigma}}^{-}(g) = \mathbb{E}_{\boldsymbol{x}\sim p(\boldsymbol{x})}[\mathbb{1}_{\hat{\sigma}(\boldsymbol{x})\leq\eta}\,\ell(-g(\boldsymbol{x}))(1-\hat{\sigma}(\boldsymbol{x}))]$$

$$+ \pi\,\mathbb{E}_{\boldsymbol{x}\sim p(\boldsymbol{x}|y=+1)}\left[\mathbb{1}_{\hat{\sigma}(\boldsymbol{x})>\eta}\,\ell(-g(\boldsymbol{x}))\frac{1-\hat{\sigma}(\boldsymbol{x})}{\hat{\sigma}(\boldsymbol{x})}\right]$$

$$+ \rho\,\mathbb{E}_{\boldsymbol{x}\sim p(\boldsymbol{x}|s=+1,y=-1)}\left[\mathbb{1}_{\hat{\sigma}(\boldsymbol{x})>\eta}\,\ell(-g(\boldsymbol{x}))\frac{1-\hat{\sigma}(\boldsymbol{x})}{\hat{\sigma}(\boldsymbol{x})}\right].$$

We notice that $\bar{R}_{s=-1,\eta,\hat{\sigma}}^{-}$ depends both on $\eta$ and $\hat{\sigma}$. It can be directly approximated from data by

$$\hat{R}_{s=-1,\eta,\hat{\sigma}}^{-}(g) = \frac{1}{n_{\mathrm{U}}}\sum_{i=1}^{n_{\mathrm{U}}}\left[\mathbb{1}_{\hat{\sigma}(\boldsymbol{x}_i^{\mathrm{U}})\leq\eta}\,\ell(-g(\boldsymbol{x}_i^{\mathrm{U}}))(1-\hat{\sigma}(\boldsymbol{x}_i^{\mathrm{U}}))\right]$$

$$+ \frac{\pi}{n_{\mathrm{P}}}\sum_{i=1}^{n_{\mathrm{P}}}\left[\mathbb{1}_{\hat{\sigma}(\boldsymbol{x}_i^{\mathrm{P}})>\eta}\,\ell(-g(\boldsymbol{x}_i^{\mathrm{P}}))\frac{1-\hat{\sigma}(\boldsymbol{x}_i^{\mathrm{P}})}{\hat{\sigma}(\boldsymbol{x}_i^{\mathrm{P}})}\right]$$

$$+ \frac{\rho}{n_{\mathrm{bN}}}\sum_{i=1}^{n_{\mathrm{bN}}}\left[\mathbb{1}_{\hat{\sigma}(\boldsymbol{x}_i^{\mathrm{bN}})>\eta}\,\ell(-g(\boldsymbol{x}_i^{\mathrm{bN}}))\frac{1-\hat{\sigma}(\boldsymbol{x}_i^{\mathrm{bN}})}{\hat{\sigma}(\boldsymbol{x}_i^{\mathrm{bN}})})\right].$$

We are now able to derive the empirical version of Equation (5) as

$$\hat{R}_{\mathrm{PUbN},\eta,\hat{\sigma}}(g) = \pi\hat{R}_{\mathrm{P}}^{+}(g) + \rho\hat{R}_{\mathrm{bN}}^{-}(g) + \hat{\bar{R}}_{s=-1,\eta,\hat{\sigma}}^{-}(g). \tag{7}$$

**Estimating $\sigma$** If we regard $s$ as a class label, the problem of estimating $\sigma$ is then equivalent to training a probabilistic classifier separating the classes with $s=+1$ and $s=-1$. Observing that $(\pi+\rho)\mathbb{E}_{\boldsymbol{x}\sim p(\boldsymbol{x}|s=+1)}[\ell(\epsilon g(x))] = \pi\mathbb{E}_{\boldsymbol{x}\sim p(\boldsymbol{x}|y=+1)}[\ell(\epsilon g(x))] + \rho\mathbb{E}_{\boldsymbol{x}\sim p(\boldsymbol{x}|y=-1,s=+1)}[\ell(\epsilon g(x))]$ for $\epsilon\in\{+1,-1\}$, it is straightforward to apply nnPU learning with availability of $\mathcal{X}_{\mathrm{P}}$, $\mathcal{X}_{\mathrm{bN}}$ and $\mathcal{X}_{\mathrm{U}}$ to

minimize $\mathbb{E}_{(\boldsymbol{x},s)\sim p(\boldsymbol{x},s)}[\ell(sg(\boldsymbol{x}))]$. In other words, here we regard $\mathcal{X}_{\mathrm{P}}$ and $\mathcal{X}_{\mathrm{bN}}$ as P and $\mathcal{X}_{\mathrm{U}}$ as U, and attempt to solve a PU learning problem by applying nnPU. Since we are interested in the class-posterior probabilities, we minimize the risk with respect to the logistic loss and apply the sigmoid function to the output of the model to get $\hat{\sigma}(\boldsymbol{x})$. However, the above risk estimator accepts any reasonable $\hat{\sigma}$ and we are not limited to using nnPU for computing $\hat{\sigma}$. For example, the least-squares fitting approach proposed in Kanamori et al. (2009) for direct density ratio estimation can also be adapted to solving the problem.

## 3.2 Estimation Error Bound

Here we establish an estimation error bound for the proposed method. Let $\mathcal{G}$ be the function class from which we find a function. The Rademacher complexity of $\mathcal{G}$ for the samples of size $n$ drawn from $q(\boldsymbol{x})$ is defined as

$$\mathfrak{R}_{n,q}(\mathcal{G}) = \mathbb{E}_{\mathcal{X}\sim q^n}\mathbb{E}_{\theta}\left[\sup_{g\in\mathcal{G}}\frac{1}{n}\sum_{x_i\in\mathcal{X}}\theta_i g(\boldsymbol{x}_i)\right],$$

where $\mathcal{X} = \{\boldsymbol{x}_1,\ldots,\boldsymbol{x}_n\}$ and $\theta = \{\theta_1,\ldots,\theta_n\}$ with each $\boldsymbol{x}_i$ drawn from $q(\boldsymbol{x})$ and $\theta_i$ as a Rademacher variable (Mohri et al., 2012). In the following we will assume that $\mathfrak{R}_{n,q}(\mathcal{G})$ vanishes asymptotically as $n \to \infty$. This holds for most of the common choices of $\mathcal{G}$ if proper regularization is considered (Bartlett & Mendelson, 2002; Golowich et al., 2018). Assume additionally the existence of $C_g > 0$ such that $\sup_{g\in\mathcal{G}}\|g\|_{\infty} \le C_g$ as well as $C_{\ell} > 0$ such that $\sup_{|z|\le C_g}\ell(z) \le C_{\ell}$. We also assume that $\ell$ is Lipschitz continuous on the interval $[-C_g, C_g]$ with a Lipschitz constant $L_{\ell}$.

**Theorem 2.** *Let* $g^* = \arg\min_{g\in\mathcal{G}} R(g)$ *be the true risk minimizer and* $\hat{g}_{\mathrm{PUbN},\eta,\hat{\sigma}} = \arg\min_{g\in\mathcal{G}} \hat{R}_{\mathrm{PUbN},\eta,\hat{\sigma}}(g)$ *be the PUbN empirical risk minimizer. We suppose that* $\hat{\sigma}$ *is a fixed function independent of data used to compute* $\hat{R}_{\mathrm{PUbN},\eta,\hat{\sigma}}(g)$ *and* $\eta \in (0,1]$. *Denote by* $p_{\mathrm{P}}(\boldsymbol{x}) = p(\boldsymbol{x} \mid y = +1)$ *and* $p_{\mathrm{bN}}(\boldsymbol{x}) = p(\boldsymbol{x} \mid y = -1, s = +1)$ *the P and bN marginals. Let* $\zeta = p(\hat{\sigma}(\boldsymbol{x}) \le \eta)$ *and* $\epsilon = \mathbb{E}_{\boldsymbol{x}\sim p(\boldsymbol{x})}[|\hat{\sigma}(\boldsymbol{x}) - \sigma(\boldsymbol{x})|^2]$. *Then for any* $\delta > 0$, *with probability at least* $1 - \delta$,

$$R(\hat{g}_{\mathrm{PUbN},\eta,\hat{\sigma}}) - R(g^*)$$

$$\le 4L_l\mathfrak{R}_{n_{\mathrm{U}},p}(\mathcal{G}) + \frac{4\pi L_l}{\eta}\mathfrak{R}_{n_{\mathrm{P}},p_{\mathrm{P}}}(\mathcal{G}) + \frac{4\rho L_l}{\eta}\mathfrak{R}_{n_{\mathrm{bN}},p_{\mathrm{bN}}}(\mathcal{G})$$

$$+ 2C_l\sqrt{\frac{\ln(6/\delta)}{2n_{\mathrm{U}}}} + \frac{2\pi C_l}{\eta}\sqrt{\frac{\ln(6/\delta)}{2n_{\mathrm{P}}}} + \frac{2\rho C_l}{\eta}\sqrt{\frac{\ln(6/\delta)}{2n_{\mathrm{bN}}}} + 2C_l\sqrt{\zeta\epsilon} + \frac{2C_l}{\eta}\sqrt{(1-\zeta)\epsilon}.$$

Theorem 2 shows that as $n_{\mathrm{P}} \to \infty$, $n_{\mathrm{bN}} \to \infty$ and $n_{\mathrm{U}} \to \infty$, we have $R(\hat{g}_{\mathrm{PUbN},\eta,\hat{\sigma}}) - R(g^*) \to 2C_l\sqrt{\zeta\epsilon} + 2(C_l/\eta)\sqrt{(1-\zeta)\epsilon}$. Furthermore, if there is $C_{\mathcal{G}} > 0$ such that $\mathfrak{R}_{n,q}(\mathcal{G}) \le C_{\mathcal{G}}/\sqrt{n}$ [2], the convergence rate is $\mathcal{O}_p(1/\sqrt{n_{\mathrm{P}}} + 1/\sqrt{n_{\mathrm{bN}}} + 1/\sqrt{n_{\mathrm{U}}})$, where $\mathcal{O}_p$ denotes the order in probability. As for $\epsilon$, knowing that $\hat{\sigma}$ is also estimated from data in practice [3], apparently its value depends on both the estimation algorithm and the number of samples that are involved in the estimation process. For example, in our approach we applied nnPU with the logistic loss to obtain $\hat{\sigma}$, so the excess risk can be written as $\mathbb{E}_{\boldsymbol{x}\sim p(\boldsymbol{x})}\mathrm{KL}(\sigma(\boldsymbol{x})\|\hat{\sigma}(\boldsymbol{x}))$, where by abuse of notation $\mathrm{KL}(p\|q) = p\ln(p/q) + (1-p)\ln((1-p)/(1-q))$ denotes the KL divergence between two Bernouilli distributions with parameters respectively $p$ and $q$. It is known that $\epsilon = \mathbb{E}_{\boldsymbol{x}\sim p(\boldsymbol{x})}[|\hat{\sigma}(\boldsymbol{x}) - \sigma(\boldsymbol{x})|^2] \le (1/2)\mathbb{E}_{\boldsymbol{x}\sim p(\boldsymbol{x})}\mathrm{KL}(\sigma(\boldsymbol{x})\|\hat{\sigma}(\boldsymbol{x}))$ (Zhang, 2004). The excess risk itself can be decomposed into the sum of the estimation error and the approximation error. Kiryo et al. (2017) showed that under mild assumptions the estimation error part converges to zero when the sample size increases to infinity in nnPU learning. It is however impossible to get rid of the approximation error part which is fixed

---

[2] For instance, this holds for linear-in-parameter model class $\mathcal{F} = \{f(\boldsymbol{x}) = \boldsymbol{w}^{\top}\phi(\boldsymbol{x}) \mid \|\boldsymbol{w}\| \le C_{\boldsymbol{w}}, \|\phi\|_{\infty} \le C_{\phi}\}$, where $C_{\boldsymbol{w}}$ and $C_{\phi}$ are positive constants (Mohri et al., 2012).

[3] These data, according to theorem 2, must be different from those used to evaluate $\hat{R}_{\mathrm{PUbN},\eta,\hat{\sigma}}(g)$. This condition is however violated in most of our experiments. See Appendix D.3 for more discussion.

once we fix the function class $\mathcal{G}$. To circumvent this problem, we can either resort to kernel-based methods with universal kernels (Zhang, 2004) or simply enlarge the function class when we get more samples.

### 3.3 PU LEARNING REVISITED

In PU learning scenarios, we only have P and U data and bN data are not available. Nevertheless, if we let $y$ play the role of $s$ and ignore all the terms related to bN data, our algorithm is naturally applicable to PU learning. Let us name the resulting algorithm PUbN\N, then

$$\hat{R}_{\text{PUbN}\backslash\text{N},\eta,\hat{\sigma}}(g) = \pi \hat{R}_{\text{P}}^+(g) + \hat{\bar{R}}_{y=-1,\eta,\hat{\sigma}}^-(g),$$

where $\hat{\sigma}$ is an estimate of $p(y = +1 \mid \boldsymbol{x})$ and

$$\bar{R}_{y=-1,\eta,\hat{\sigma}}^-(g) = \mathbb{E}_{\boldsymbol{x}\sim p(\boldsymbol{x})}[\mathbb{1}_{\hat{\sigma}(\boldsymbol{x})\leq\eta}\,\ell(-g(\boldsymbol{x}))(1-\hat{\sigma}(\boldsymbol{x}))] + \pi\,\mathbb{E}_{\boldsymbol{x}\sim p(\boldsymbol{x}|y=+1)}\left[\mathbb{1}_{\hat{\sigma}(\boldsymbol{x})>\eta}\,\ell(-g(\boldsymbol{x}))\tfrac{1-\hat{\sigma}(\boldsymbol{x})}{\hat{\sigma}(\boldsymbol{x})}\right].$$

PUbN\N can be viewed as a variant of the traditional two-step approach in PU learning which first identifies possible N data in U data and then perform ordinary PN classification to distinguish P data from the identified N data. However, being based on state-of-the-art nnPU learning, our method is more promising than other similar algorithms. Moreover, by explicitly considering the posterior $p(y = +1 \mid \boldsymbol{x})$, we attempt to correct the bias induced by the fact of only taking into account confident negative samples. The benefit of using an unbiased risk estimator is that the resulting algorithm is always statistically consistent, i.e., the estimation error converges in probability to zero as the number of samples grows to infinity.

## 4 EXPERIMENTS

In this section, we experimentally investigate the proposed method and compare its performance against several baseline methods.

### 4.1 BASIC SETUP

We focus on training neural networks with stochastic optimization. For simplicity, in an experiment, $\hat{\sigma}$ and $g$ always use the same model and are trained for the same number of epochs. All models are learned using AMSGrad (Reddi et al., 2018) as the optimizer and the logistic loss as the surrogate loss unless otherwise specified. To determine the value of $\eta$, we introduce another hyperparameter $\tau$ and choose $\eta$ such that $\#\{x \in \mathcal{X}_\text{U} \mid \hat{\sigma}(x) \leq \eta\} = \tau(1 - \pi - \rho)n_\text{U}$. In all the experiments, an additional validation set, equally composed of P, U and bN data, is sampled for both hyperparameter tuning and choosing the model parameters with the lowest validation loss among those obtained after every epoch. Regarding the computation of the validation loss, we use the PU risk estimator (2) with the sigmoid loss for $g$ and an empirical approximation of $\mathbb{E}_{\boldsymbol{x}\sim p(\boldsymbol{x})}[|\hat{\sigma}(\boldsymbol{x})-\sigma(\boldsymbol{x})|^2]-\mathbb{E}_{\boldsymbol{x}\sim p(\boldsymbol{x})}[\sigma(\boldsymbol{x})^2]$ for $\hat{\sigma}$ (see Appendix B).

### 4.2 EFFECTIVENESS OF THE ALGORITHM

We assess the performance of the proposed method on three benchmark datasets: MNIST, CIFAR-10 and 20 Newsgroups. Experimental details are given in Appendix C. In particular, since all the three datasets are originally designed for multiclass classification, we group different categories together to form a binary classification problem.

**Baselines.** When $\mathcal{X}_\text{bN}$ is given, two baseline methods are considered. The first one is nnPNU adapted from (4). In the second method, named as PU→PN, we train two binary classifiers: one is learned with nnPU while we regard $s$ as the class label, and the other is learned from $\mathcal{X}_\text{P}$ and $\mathcal{X}_\text{bN}$ to separate P samples from bN samples. A sample is classified in the P class only if it is so classified by the two classifiers. When $\mathcal{X}_\text{bN}$ is not available, nnPU is compared with the proposed PUbN\N.

**Sampling bN Data** To sample $\mathcal{X}_\text{bN}$, we suppose that the bias of N data is caused by a latent prior probability change (Sugiyama & Storkey, 2007; Hu et al., 2018) in the N class. Let $z \in \mathcal{Z} :=$

Table 1: Mean and standard deviation of misclassification rates over 10 trials for MNIST, CIFAR-10 and 20 Newsgroups under different choices of P class and bN data sampling strategies. For a same learning task, different methods are compared using the same 10 random samplings. Underlines denote that with the use of bN data the method leads to an improvement of performance according to the 5% t-test. Boldface indicates the best method in each task.
[†] Biased N data uniformly sampled from the indicated latent categories.
[*] Probabilities that a sample of $\mathcal{X}_{bN}$ belongs to the latent categories [1, 3, 5, 7, 9] / [bird, cat, deer, dog, frog, horse] / [sci., soc., talk.] are [0.03, 0.15, 0.3, 0.02, 0.5] / [0.1, 0.02, 0.2, 0.08, 0.2, 0.4] / [0.1, 0.5, 0.4].

| Dataset | P | biased N | $\rho$ | nnPU/nnPNU | PUbN(\N) | PU→PN |
|---|---|---|---|---|---|---|
| MNIST | 2, 4, 6, 8, 10 | Not given | NA | $5.76 \pm 1.04$ | $\mathbf{4.64 \pm 0.62}$ | NA |
| | | 1, 3, 5 [†] | 0.3 | $5.33 \pm 0.97$ | $\underline{4.05 \pm 0.27}$ | $\mathbf{\underline{4.00 \pm 0.30}}$ |
| | | 9 > 5 > others [*] | 0.2 | $4.60 \pm 0.65$ | $\underline{3.91 \pm 0.66}$ | $\mathbf{\underline{3.77 \pm 0.31}}$ |
| CIFAR-10 | Airplane, automobile, ship, truck | Not given | NA | $12.02 \pm 0.65$ | $\mathbf{10.70 \pm 0.57}$ | NA |
| | | Cat, dog, horse [†] | 0.3 | $10.25 \pm 0.38$ | $\mathbf{\underline{9.71 \pm 0.51}}$ | $10.37 \pm 0.65$ |
| | | Horse > deer = frog > others [*] | 0.25 | $\underline{9.98 \pm 0.53}$ | $\mathbf{9.92 \pm 0.42}$ | $\underline{10.17 \pm 0.35}$ |
| CIFAR-10 | Cat, deer, dog, horse | Not given | NA | $23.78 \pm 1.04$ | $\mathbf{21.13 \pm 0.90}$ | NA |
| | | Bird, frog [†] | 0.2 | $22.00 \pm 0.53$ | $\mathbf{\underline{18.83 \pm 0.71}}$ | $\underline{19.88 \pm 0.62}$ |
| | | Car, truck [†] | 0.2 | $22.00 \pm 0.74$ | $\mathbf{\underline{20.19 \pm 1.06}}$ | $21.83 \pm 1.36$ |
| 20 Newsgroups | alt., comp., misc., rec. | Not given | NA | $14.67 \pm 0.87$ | $\mathbf{13.30 \pm 0.53}$ | NA |
| | | sci.[†] | 0.21 | $14.69 \pm 0.46$ | $\mathbf{13.10 \pm 0.90}$ | $13.58 \pm 0.97$ |
| | | talk.[†] | 0.17 | $14.38 \pm 0.74$ | $\mathbf{\underline{12.61 \pm 0.75}}$ | $13.76 \pm 0.66$ |
| | | soc. > talk. > sci.[*] | 0.1 | $14.41 \pm 0.76$ | $\mathbf{\underline{12.18 \pm 0.59}}$ | $12.92 \pm 0.51$ |

$\{1, \ldots, S\}$ be some latent variable which we call a latent category, where $S$ is a constant. It is assumed

$$p(\boldsymbol{x} \mid z, y = -1) = p(\boldsymbol{x} \mid z, y = -1, s = +1),$$
$$p(z \mid y = -1) \neq p(z \mid y = -1, s = +1).$$

In the experiments, the latent categories are the original class labels of the datasets. Concrete definitions of $\mathcal{X}_{bN}$ with experimental results are summarized in Table 1.

**Results.** Overall, our proposed method consistently achieves the best or comparable performance in all the scenarios, including those of standard PU learning. Additionally, using bN data can effectively help improving classification performance. However, the choice of algorithm is essential. Both nnPNU and the naive PU→PN are able to leverage bN data to enhance classification accuracy in only relatively few tasks. In the contrast, the proposed PUbN successfully reduce the misclassification error most of the time.

Clearly, the performance gain that we can benefit from the availability of bN data is case-dependent. On CIFAR-10, the greatest improvement is achieved when we regard mammals (i.e. cat, deer, dog and horse) as P class and drawn samples from latent categories bird and frog as labeled negative data. This is not surprising because birds and frogs are more similar to mammals than vehicles, which makes the classification harder specifically for samples from these two latent categories. By explicitly labeling these samples as N data, we allow the classifier to make better predictions for these difficult samples.

### 4.3 The Presence of bN Data Helps: An Illustration

Through experiments we have demonstrated that the presence of bN data effectively helps learning a better classifier. Here we would like to provide some intuition for the reason behind this. Let us consider the MNIST learning task where $\mathcal{X}_{bN}$ is uniformly sampled from the latent categories 1, 3 and 5. We project the representations learned by the classifier (i.e., the activation values of the last layer of the neural network) into a 2D plane using PCA for both nnPU and PUbN algorithms.

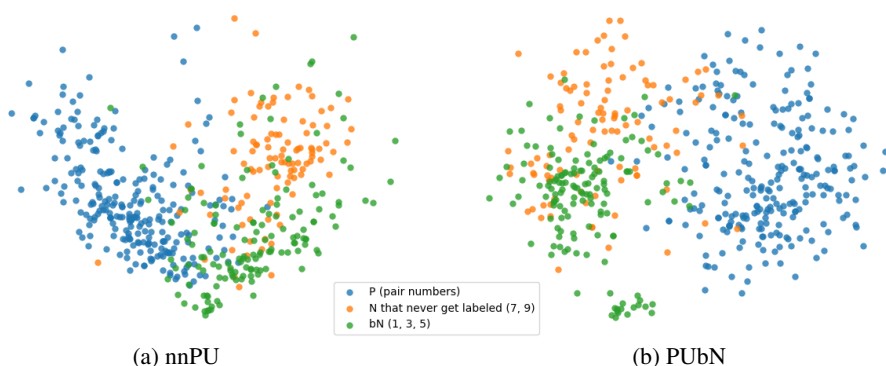

(a) nnPU                                                    (b) PUbN

Figure 1: PCA embeddings of the representations learned by the nnPU and PUbN classifiers for 500 samples from the test set in the MNIST learning task where $\mathcal{X}_{\text{bn}}$ is uniformly sampled from latent categories 1, 3 and 5.

The results are shown in Figure 1. Since for both nnPU and PUbN classifiers, the first two principal components account around 90% of variance, we believe that this figure depicts fairly well the learned representations. Thanks to the use of bN data, in the high-level feature space 1, 3, 5 and P data are further pushed away when we employ the proposed PUbN learning algorithm, and we are always able to separate 7, 9 from P to some extent. This explains the better performance which is achieved by PUbN learning and the benefit of incorporating bN data into the learning process.

## 5 CONCLUSION

This paper studied the PUbN classification problem, where a binary classifier is trained on P, U and bN data. The proposed method is a two-step approach inspired from both PU learning and importance weighting. The key idea is to attribute appropriate weights to each example to evaluate the classification risk using the three sets of data. We theoretically established an estimation error bound for the proposed risk estimator and experimentally showed that our approach successfully leveraged bN data to improve the classification performance on several real-world datasets. A variant of our algorithm was able to achieve state-of-the-art results in PU learning.

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

APPENDIX

## A PROOFS

### A.1 PROOF OF THEOREM 1

We notice that $(1 - \pi - \rho)p(\boldsymbol{x} \mid s = -1) = p(\boldsymbol{x}, s = -1)$ and that when $h(\boldsymbol{x}) > \eta$, we have $p(s = +1 \mid \boldsymbol{x}) = \sigma(\boldsymbol{x}) > 0$, which allows us to write $p(s = -1 \mid \boldsymbol{x}) = (p(s = -1 \mid \boldsymbol{x})/p(s = +1 \mid \boldsymbol{x}))p(s = +1 \mid \boldsymbol{x})$. We can thus decompose $\bar{R}^-_{s=-1}(g)$ as following:

$$
\begin{aligned}
\bar{R}^-_{s=-1}(g) &= \int \ell(-g(\boldsymbol{x}))p(\boldsymbol{x}, s = -1)\, dx \\
&= \int \mathbb{1}_{h(\boldsymbol{x}) \leq \eta}\, \ell(-g(\boldsymbol{x}))p(\boldsymbol{x}, s = -1)\, dx \\
&\quad + \int \mathbb{1}_{h(\boldsymbol{x}) > \eta}\, \ell(-g(\boldsymbol{x}))p(\boldsymbol{x}, s = -1)\, dx \\
&= \int \mathbb{1}_{h(\boldsymbol{x}) \leq \eta}\, \ell(-g(\boldsymbol{x})) \frac{p(\boldsymbol{x}, s = -1)}{p(\boldsymbol{x})} p(\boldsymbol{x})\, dx \\
&\quad + \int \mathbb{1}_{h(\boldsymbol{x}) > \eta}\, \ell(-g(\boldsymbol{x})) \frac{p(\boldsymbol{x}, s = -1)}{p(\boldsymbol{x}, s = +1)} p(\boldsymbol{x}, s = +1)\, dx.
\end{aligned}
$$

By writing $p(\boldsymbol{x}, s = -1) = p(s = -1 \mid \boldsymbol{x})p(\boldsymbol{x}) = (1 - \sigma(\boldsymbol{x}))p(\boldsymbol{x})$ and $p(\boldsymbol{x}, s = +1) = p(s = +1 \mid \boldsymbol{x})p(\boldsymbol{x}) = \sigma(\boldsymbol{x})p(\boldsymbol{x})$, we have

$$
\begin{aligned}
\bar{R}^-_{s=-1}(g) &= \int \mathbb{1}_{h(\boldsymbol{x}) \leq \eta}\, \ell(-g(\boldsymbol{x}))(1 - \sigma(\boldsymbol{x}))p(\boldsymbol{x})\, dx \\
&\quad + \int \mathbb{1}_{h(\boldsymbol{x}) > \eta}\, \ell(-g(\boldsymbol{x})) \frac{1 - \sigma(\boldsymbol{x})}{\sigma(\boldsymbol{x})} p(\boldsymbol{x}, s = +1)\, dx.
\end{aligned}
$$

We obtain Equation (6) after replacing $p(\boldsymbol{x}, s = +1)$ by $\pi p(x \mid y = +1) + \rho p(x \mid y = -1, s = +1)$.

### A.2 PROOF OF THEOREM 2

For $\hat{\sigma}$ and $\eta$ given, let us define

$$
R_{\text{PUbN}, \eta, \hat{\sigma}}(g) = \pi R^+_{\text{P}}(g) + \rho R^-_{\text{bN}}(g) + \bar{R}^-_{s=-1, \eta, \hat{\sigma}}(g).
$$

The following lemma establishes the uniform deviation bound from $\hat{R}_{\text{PUbN}, \eta, \hat{\sigma}}$ to $R_{\text{PUbN}, \eta, \hat{\sigma}}$.

**Lemma 1.** *Let $\hat{\sigma} : \mathbb{R}^d \to [0, 1]$ be a fixed function independent of data used to compute $\hat{R}_{\text{PUbN}, \eta, \hat{\sigma}}$ and $\eta \in (0, 1]$. For any $\delta > 0$, with probability at least $1 - \delta$,*

$$
\begin{aligned}
\sup_{g \in \mathcal{G}} &|\hat{R}^-_{\text{PUbN}, \eta, \hat{\sigma}}(g) - R_{\text{PUbN}, \eta, \hat{\sigma}}(g)| \\
&\leq 2L_l \mathfrak{R}_{n_{\text{U}}, p}(\mathcal{G}) + \frac{2\pi L_l}{\eta} \mathfrak{R}_{n_{\text{P}}, p_{\text{P}}}(\mathcal{G}) + \frac{2\rho L_l}{\eta} \mathfrak{R}_{n_{\text{bN}}, p_{\text{bN}}}(\mathcal{G}) \\
&\quad + C_l \sqrt{\frac{\ln(6/\delta)}{2n_{\text{U}}}} + \frac{\pi C_l}{\eta} \sqrt{\frac{\ln(6/\delta)}{2n_{\text{P}}}} + \frac{\rho C_l}{\eta} \sqrt{\frac{\ln(6/\delta)}{2n_{\text{bN}}}}.
\end{aligned}
$$

*Proof.* For ease of notation, let

$$R_{\mathrm{P}}(g) = \mathbb{E}_{\boldsymbol{x} \sim p_{\mathrm{P}}(\boldsymbol{x})} \left[ \ell(g(\boldsymbol{x})) + \mathbb{1}_{\hat{\sigma}(\boldsymbol{x}) > \eta} \, \ell(-g(\boldsymbol{x})) \frac{1 - \hat{\sigma}(\boldsymbol{x})}{\hat{\sigma}(\boldsymbol{x})} \right],$$

$$R_{\mathrm{bN}}(g) = \mathbb{E}_{\boldsymbol{x} \sim p_{\mathrm{bN}}(\boldsymbol{x})} \left[ \ell(-g(\boldsymbol{x}))(1 + \mathbb{1}_{\hat{\sigma}(\boldsymbol{x}) > \eta} \frac{1 - \hat{\sigma}(\boldsymbol{x})}{\hat{\sigma}(\boldsymbol{x})}) \right],$$

$$R_{\mathrm{U}}(g) = \mathbb{E}_{\boldsymbol{x} \sim p(\boldsymbol{x})} \left[ \mathbb{1}_{\hat{\sigma}(\boldsymbol{x}) \leq \eta} \, \ell(-g(\boldsymbol{x}))(1 - \hat{\sigma}(\boldsymbol{x})) \right],$$

$$\hat{R}_{\mathrm{P}}(g) = \frac{1}{n_{\mathrm{P}}} \sum_{i=1}^{n_{\mathrm{P}}} \left[ \ell(g(\boldsymbol{x}_i^{\mathrm{P}})) + \mathbb{1}_{\hat{\sigma}(\boldsymbol{x}_i^{\mathrm{P}}) > \eta} \, \ell(-g(\boldsymbol{x}_i^{\mathrm{P}})) \frac{1 - \hat{\sigma}(\boldsymbol{x}_i^{\mathrm{P}})}{\hat{\sigma}(\boldsymbol{x}_i^{\mathrm{P}})} \right],$$

$$\hat{R}_{\mathrm{bN}}(g) = \frac{1}{n_{\mathrm{bN}}} \sum_{i=1}^{n_{\mathrm{bN}}} \left[ \ell(-g(\boldsymbol{x}_i^{\mathrm{bN}}))(1 + \mathbb{1}_{\hat{\sigma}(\boldsymbol{x}_i^{\mathrm{bN}}) > \eta} \frac{1 - \hat{\sigma}(\boldsymbol{x}_i^{\mathrm{bN}})}{\hat{\sigma}(\boldsymbol{x}_i^{\mathrm{bN}})}) \right],$$

$$\hat{R}_{\mathrm{U}}(g) = \frac{1}{n_{\mathrm{U}}} \sum_{i=1}^{n_{\mathrm{U}}} \left[ \mathbb{1}_{\hat{\sigma}(\boldsymbol{x}_i^{\mathrm{U}}) \leq \eta} \, \ell(-g(\boldsymbol{x}_i^{\mathrm{U}}))(1 - \hat{\sigma}(\boldsymbol{x}_i^{\mathrm{U}})) \right].$$

From the sub-additivity of the supremum operator, we have

$$\sup_{g \in \mathcal{G}} |\hat{R}_{\mathrm{PUbN}, \eta, \hat{\sigma}}^-(g) - R_{\mathrm{PUbN}, \eta, \hat{\sigma}}(g)|$$

$$\leq \pi \sup_{g \in \mathcal{G}} |\hat{R}_{\mathrm{P}}(g) - R_{\mathrm{P}}(g)| + \rho \sup_{g \in \mathcal{G}} |\hat{R}_{\mathrm{bN}}(g) - R_{\mathrm{bN}}(g)| + \sup_{g \in \mathcal{G}} |\hat{R}_{\mathrm{U}}(g) - R_{\mathrm{U}}(g)|.$$

As a consequence, to conclude the proof, it suffices to prove that with probability at least $1 - \delta/3$, the following bounds hold separately:

$$\sup_{g \in \mathcal{G}} |\hat{R}_{\mathrm{P}}(g) - R_{\mathrm{P}}(g)| \leq \frac{2L_l}{\eta} \mathfrak{R}_{n_{\mathrm{P}}, p_{\mathrm{P}}}(\mathcal{G}) + \frac{C_l}{\eta} \sqrt{\frac{\ln(6/\delta)}{2n_{\mathrm{P}}}}, \tag{8}$$

$$\sup_{g \in \mathcal{G}} |\hat{R}_{\mathrm{bN}}(g) - R_{\mathrm{bN}}(g)| \leq \frac{2L_l}{\eta} \mathfrak{R}_{n_{\mathrm{bN}}, p_{\mathrm{bN}}}(\mathcal{G}) + \frac{C_l}{\eta} \sqrt{\frac{\ln(6/\delta)}{2n_{\mathrm{bN}}}}, \tag{9}$$

$$\sup_{g \in \mathcal{G}} |\hat{R}_{\mathrm{U}}(g) - R_{\mathrm{U}}(g)| \leq 2L_l \mathfrak{R}_{n_{\mathrm{U}}, p}(\mathcal{G}) + C_l \sqrt{\frac{\ln(6/\delta)}{2n_{\mathrm{U}}}}. \tag{10}$$

Below we prove (8). (9) and (10) are proven similarly.

Let $\phi_{\boldsymbol{x}} : \mathbb{R} \to \mathbb{R}_+$ be the function defined by $\phi_{\boldsymbol{x}} : z \mapsto \ell(z) + \mathbb{1}_{\hat{\sigma}(\boldsymbol{x}) > \eta} \, \ell(-z)((1 - \hat{\sigma}(\boldsymbol{x}))/\hat{\sigma}(\boldsymbol{x}))$. For $\boldsymbol{x} \in \mathbb{R}^d, g \in \mathcal{G}$, since $\ell(g(\boldsymbol{x})) \in [0, C_l]$, $\ell(-g(\boldsymbol{x})) \in [0, C_l]$ and $\mathbb{1}_{\hat{\sigma}(\boldsymbol{x}) > \eta}((1 - \hat{\sigma}(\boldsymbol{x}))/\hat{\sigma}(\boldsymbol{x})) \in [0, (1 - \eta)/\eta]$, we always have $\phi_{\boldsymbol{x}}(g(\boldsymbol{x})) \in [0, C_l/\eta]$. Following the proof of Theorem 3.1 in Mohri et al. (2012), it is then straightforward to show that with probability at least $1 - \delta/3$, it holds that

$$\sup_{g \in \mathcal{G}} |\hat{R}_{\mathrm{P}}(g) - R_{\mathrm{P}}(g)| \leq 2 \, \mathbb{E}_{\mathcal{X}_{\mathrm{P}} \sim p_{\mathrm{P}}^{n_{\mathrm{P}}}} \mathbb{E}_\theta \left[ \sup_{g \in \mathcal{G}} \frac{1}{n_{\mathrm{P}}} \sum_{i=1}^{n_{\mathrm{P}}} \theta_i \phi_{\boldsymbol{x}_i}(g(\boldsymbol{x}_i)) \right] + \frac{C_l}{\eta} \sqrt{\frac{\ln(6/\delta)}{2n_{\mathrm{P}}}},$$

where $\theta = \{\theta_1, \ldots, \theta_{n_{\mathrm{P}}}\}$ and each $\theta_i$ is a Rademacher variable.

Also notice that for all $\boldsymbol{x}$, $\phi_{\boldsymbol{x}}$ is a $(L_l/\eta)$-Lipschitz function on the interval $[-C_g, C_g]$. By using a modified version of Talagrad's concentration lemma (specifically, Lemma 26.9 in Shalev-Shwartz & Ben-David (2014)), we can show that, when the set $\mathcal{X}_{\mathrm{P}}$ is fixed, we have

$$\mathbb{E}_\theta \left[ \sup_{g \in \mathcal{G}} \frac{1}{n_{\mathrm{P}}} \sum_{i=1}^{n_{\mathrm{P}}} \theta_i \phi_{\boldsymbol{x}_i}(g(\boldsymbol{x}_i)) \right] \leq \frac{L_l}{\eta} \mathbb{E}_\theta \left[ \sup_{g \in \mathcal{G}} \frac{1}{n_{\mathrm{P}}} \sum_{i=1}^{n_{\mathrm{P}}} \theta_i g(\boldsymbol{x}_i) \right].$$

After taking expectation over $\mathcal{X}_{\mathrm{P}} \sim p_{\mathrm{P}}^{n_{\mathrm{P}}}$, we obtain the Equation (8). $\qquad \square$

However, what we really want to minimize is the true risk $R(g)$. Therefore, we also need to bound the difference between $R_{\mathrm{PUbN},\eta,\hat{\sigma}}(g)$ and $R(g)$, or equivalently, the difference between $\bar{R}^-_{s=-1,\eta,\hat{\sigma}}(g)$ and $\bar{R}^-_{s=-1}(g)$.

**Lemma 2.** *Let $\hat{\sigma} : \mathbb{R}^d \to [0,1]$, $\eta \in (0,1]$, $\zeta = p(\hat{\sigma} \leq \eta)$ and $\epsilon = \mathbb{E}_{\boldsymbol{x} \sim p(\boldsymbol{x})}[|\hat{\sigma}(\boldsymbol{x}) - \sigma(\boldsymbol{x})|^2]$. For all $g \in \mathcal{G}$, it holds that*

$$|\bar{R}^-_{s=-1,\eta,\hat{\sigma}}(g) - \bar{R}^-_{s=-1}(g)| \leq C_l \sqrt{\zeta\epsilon} + \frac{C_l}{\eta}\sqrt{(1-\zeta)\epsilon}.$$

*Proof.* One one hand, we have

$$\bar{R}^-_{s=-1}(g) = \underbrace{\int \mathbb{1}_{\hat{\sigma}(\boldsymbol{x})\leq\eta}\,\ell(-g(\boldsymbol{x}))(1-\sigma(\boldsymbol{x}))p(\boldsymbol{x})\,dx}_{A_1}$$

$$+ \underbrace{\int \mathbb{1}_{\hat{\sigma}(\boldsymbol{x})>\eta}\,\ell(-g(\boldsymbol{x}))(1-\sigma(\boldsymbol{x}))p(\boldsymbol{x})\,dx}_{B_1}.$$

On the other hand, we can express $\bar{R}^-_{s=-1,\eta,\hat{\sigma}}(g)$ as

$$\bar{R}^-_{s=-1,\eta,\hat{\sigma}}(g) = \int \mathbb{1}_{\hat{\sigma}(\boldsymbol{x})\leq\eta}\,\ell(-g(\boldsymbol{x}))(1-\hat{\sigma}(\boldsymbol{x}))p(\boldsymbol{x})\,dx$$

$$+ \int \mathbb{1}_{\hat{\sigma}(\boldsymbol{x})>\eta}\,\ell(-g(\boldsymbol{x}))\frac{1-\hat{\sigma}(\boldsymbol{x})}{\hat{\sigma}(x)}p(\boldsymbol{x}, s = +1)\,dx.$$

$$= \underbrace{\int \mathbb{1}_{\hat{\sigma}(\boldsymbol{x})\leq\eta}\,\ell(-g(\boldsymbol{x}))(1-\hat{\sigma}(\boldsymbol{x}))p(\boldsymbol{x})\,dx}_{A_2}$$

$$+ \underbrace{\int \mathbb{1}_{\hat{\sigma}(\boldsymbol{x})>\eta}\,\ell(-g(\boldsymbol{x}))(1-\hat{\sigma}(\boldsymbol{x}))\frac{\sigma(\boldsymbol{x})}{\hat{\sigma}(\boldsymbol{x})}p(\boldsymbol{x})\,dx}_{B_2}.$$

The last equality follows from the fact $p(\boldsymbol{x}, s = +1) = \sigma(\boldsymbol{x})p(\boldsymbol{x})$. As $|\bar{R}^-_{s=-1,\eta,\hat{\sigma}}(g) - \bar{R}^-_{s=-1}(g)| \leq |A_1 - A_2| + |B_1 - B_2|$, it is sufficient to derive bounds for $|A_1 - A_2|$ and $|B_1 - B_2|$ separately. For $|B_1 - B_2|$, we write

$$|B_1 - B_2| \leq \int \mathbb{1}_{\hat{\sigma}(\boldsymbol{x})>\eta}\,\ell(-g(\boldsymbol{x}))\frac{|\hat{\sigma}(\boldsymbol{x}) - \sigma(\boldsymbol{x})|}{\hat{\sigma}(\boldsymbol{x})}p(\boldsymbol{x})\,dx$$

$$\leq \frac{C_l}{\eta}\int \mathbb{1}_{\hat{\sigma}(\boldsymbol{x})>\eta}|\hat{\sigma}(\boldsymbol{x}) - \sigma(\boldsymbol{x})|p(\boldsymbol{x})\,dx$$

$$\leq \frac{C_l}{\eta}\left(\int \mathbb{1}^2_{\hat{\sigma}(\boldsymbol{x})>\eta}p(\boldsymbol{x})\,dx\right)^{\frac{1}{2}}\left(\int |\hat{\sigma}(\boldsymbol{x}) - \sigma(\boldsymbol{x})|^2p(\boldsymbol{x})\,dx\right)^{\frac{1}{2}}$$

$$= \frac{C_l}{\eta}\sqrt{(1-\zeta)\epsilon}$$

From the second to the third line we use the Cauchy-Schwarz inequality. $|A_1 - A_2| \leq C_l\sqrt{\zeta\epsilon}$ can be proven similarly, which concludes the proof. $\square$

Combining lemma 1 and lemma 2, we know that with probability at least $1 - \delta$, the following holds:

$$\sup_{g \in \mathcal{G}} |\hat{R}^-_{\text{PUbN},\eta,\hat{\sigma}}(g) - R(g)|$$

$$\leq 2L_l \mathfrak{R}_{n_{\text{U}},p}(\mathcal{G}) + \frac{2\pi L_l}{\eta} \mathfrak{R}_{n_{\text{P}},p_{\text{P}}}(\mathcal{G}) + \frac{2\rho L_l}{\eta} \mathfrak{R}_{n_{\text{bN}},p_{\text{bN}}}(\mathcal{G})$$

$$+ C_l \sqrt{\frac{\ln(6/\delta)}{2n_{\text{U}}}} + \frac{\pi C_l}{\eta} \sqrt{\frac{\ln(6/\delta)}{2n_{\text{P}}}} + \frac{\rho C_l}{\eta} \sqrt{\frac{\ln(6/\delta)}{2n_{\text{bN}}}} + C_l \sqrt{\zeta \epsilon} + \frac{C_l}{\eta} \sqrt{(1-\zeta)\epsilon}.$$

Finally, with probability at least $1 - \delta$,

$$R(\hat{g}_{\text{PUbN},\eta,\hat{\sigma}}) - R(g^*)$$

$$= (R(\hat{g}_{\text{PUbN},\eta,\hat{\sigma}}) - \hat{R}^-_{\text{PUbN},\eta,\hat{\sigma}}(\hat{g}_{\text{PUbN},\eta,\hat{\sigma}}))$$

$$+ (\hat{R}^-_{\text{PUbN},\eta,\hat{\sigma}}(\hat{g}_{\text{PUbN},\eta,\hat{\sigma}}) - \hat{R}^-_{\text{PUbN},\eta,\hat{\sigma}}(g^*)) + (\hat{R}^-_{\text{PUbN},\eta,\hat{\sigma}}(g^*) - R(g^*))$$

$$\leq \sup_{g \in \mathcal{G}} |\hat{R}^-_{\text{PUbN},\eta,\hat{\sigma}}(g) - R(g)| + 0 + \sup_{g \in \mathcal{G}} |\hat{R}^-_{\text{PUbN},\eta,\hat{\sigma}}(g) - R(g)|$$

$$\leq 4L_l \mathfrak{R}_{n_{\text{U}},p}(\mathcal{G}) + \frac{4\pi L_l}{\eta} \mathfrak{R}_{n_{\text{P}},p_{\text{P}}}(\mathcal{G}) + \frac{4\rho L_l}{\eta} \mathfrak{R}_{n_{\text{bN}},p_{\text{bN}}}(\mathcal{G})$$

$$+ 2C_l \sqrt{\frac{\ln(6/\delta)}{2n_{\text{U}}}} + \frac{2\pi C_l}{\eta} \sqrt{\frac{\ln(6/\delta)}{2n_{\text{P}}}} + \frac{2\rho C_l}{\eta} \sqrt{\frac{\ln(6/\delta)}{2n_{\text{bN}}}} + 2C_l \sqrt{\zeta \epsilon} + \frac{2C_l}{\eta} \sqrt{(1-\zeta)\epsilon}.$$

The first inequality uses the definition of $\hat{g}_{\text{PUbN},\eta,\hat{\sigma}}$.

## B  VALIDATION LOSS FOR ESTIMATION OF $\sigma$

In terms of validation we want to choose the model for $\hat{\sigma}$ such that $J_0(\hat{\sigma}) = \mathbb{E}_{\boldsymbol{x} \sim p(\boldsymbol{x})}[|\hat{\sigma}(\boldsymbol{x}) - \sigma(\boldsymbol{x})|^2]$ is minimized. Since $\sigma(\boldsymbol{x})p(\boldsymbol{x}) = p(\boldsymbol{x}, s = +1)$, we have

$$J_0(\hat{\sigma}) = \int (\hat{\sigma}(\boldsymbol{x}) - \sigma(\boldsymbol{x}))^2 p(\boldsymbol{x}) \, dx$$

$$= \int \hat{\sigma}(\boldsymbol{x})^2 p(\boldsymbol{x}) \, dx - 2 \int \hat{\sigma}(\boldsymbol{x}) p(\boldsymbol{x}, s = +1) \, dx + \int \sigma(\boldsymbol{x})^2 p(\boldsymbol{x}) \, dx.$$

The last term does not depend on $\hat{\sigma}$ and can be ignored if we want to identify $\hat{\sigma}$ achieving the smallest $J(\hat{\sigma})$. We denote by $J(\hat{\sigma})$ the sum of the first two terms. The middle term can be further expanded using

$$\int \hat{\sigma}(\boldsymbol{x}) p(\boldsymbol{x}, s = +1) \, dx = \pi \int \hat{\sigma}(\boldsymbol{x}) p(\boldsymbol{x} \mid y = +1) \, dx + \rho \int \hat{\sigma}(\boldsymbol{x}) p(\boldsymbol{x} \mid y = -1, s = +1) \, dx.$$

The validation loss of an estimation $\hat{\sigma}$ is then defined as

$$\hat{J}(\hat{\sigma}) = \frac{1}{n_{\text{U}}} \sum_{i=1}^{n_{\text{U}}} \hat{\sigma}(\boldsymbol{x}_i^{\text{U}})^2 - \frac{2\pi}{n_{\text{P}}} \sum_{i=1}^{n_{\text{P}}} \hat{\sigma}(\boldsymbol{x}_i^{\text{P}}) - \frac{2\rho}{n_{\text{bN}}} \sum_{i=1}^{n_{\text{bN}}} \hat{\sigma}(\boldsymbol{x}_i^{\text{bN}}).$$

It is also possible to minimize this value directly to acquire $\hat{\sigma}$. In our experiments we decide to learn $\hat{\sigma}$ by nnPU for a better comparison between different methods.

## C  DETAILED EXPERIMENTAL SETTING

### C.1  FROM MULTICLASS TO BINARY CLASS

In the experiments we work on multiclass classification datasets. Therefore it is necessary to define the P and N classes ourselves. MNIST is processed in such a way that pair numbers 0, 2, 4, 6, 8 form

the P class and impair numbers 1, 3, 5, 7, 9 form the N class. Accordingly, $\pi = 0.49$. For CIFAR-10, we consider two definitions of the P class. The first one corresponds to a quite natural task that aims to distinguish vehicles from animals. Airplane, automobile, ship and truck are therefore defined to be the P class while the N class is formed by bird, cat, deer, dog, frog and horse. For the sake of diversity, we also study another task in which we attempt to distinguish the mammals from the non-mammals. The P class is then formed by cat, deer, dog, and horse while the N class consists of the other six classes. We have $\pi = 0.4$ in the two cases. As for 20 Newsgroups, alt., comp., misc. and rec. make up the P class whereas sci., soc. and talk. make up the N class. This gives $\pi = 0.56$.

## C.2 Training, Validation and Test Set

For the three datasets, we use the standard test examples as a held-out test set. The test set size is thus of 10000 for MNIST and CIFAR-10, and 7528 for 20 Newsgroups. Regarding the training set, we sample 500, 500 and 6000 P, bN and U training examples for MNIST and 20 Newsgroups, and 1000, 1000 and 10000 P, bN and U training examples for CIFAR-10. The validation set is always five times smaller than the training set.

## C.3 20 Newsgroups Preprocessing

The original 20 Newsgroups dataset contains raw text data and needs to be preprocessed into text feature vectors for classification. In our experiments we borrow the pre-trained ELMo word embedding (Peters et al., 2018) from https://allennlp.org/elmo. The used 5.5B model was, according to the website, trained on a dataset of 5.5B tokens consisting of Wikipedia (1.9B) and all of the monolingual news crawl data from WMT 2008-2012 (3.6B). For each word, we concatenate the features from the three layers of the ELMo model, and for each document, as suggested in Rücklé et al. (2018), we concatenate the average, minimum, and maximum computed along the word dimension. This results in a 9216-dimensional feature vector for a single document.

## C.4 Models and Hyperparameters

**MNIST** For MNIST, we use a standard ConvNet with ReLU. This model contains two 5x5 convolutional layers and one fully-connected layer, with each convolutional layer followed by a 2x2 max pooling. The channel sizes are 5-10-40. The model is trained for 100 epochs with a weight decay of $10^{-4}$. Each minibatch is made up of 10 P, 10 bN (if available) and 120 U samples. The learning rate $\alpha \in \{10^{-2}, 10^{-3}\}$ and $\tau \in \{0.5, 0.7, 0.9\}$, $\gamma \in \{0.1, 0.3, 0.5, 0.7, 0.9\}$ are selected with validation data.

**CIFAR-10** For CIFAR-10, we train PreAct ResNet-18 (He et al., 2016) for 200 epochs and the learning rate is divided by 10 after 80 epochs and 120 epochs. This is a common practice and similar adjustment can be found in He et al. (2016). The weight decay is set to $10^{-4}$. The minibatch size is 1/100 of the number of training samples, and the initial learning rate is chosen from $\{10^{-2}, 10^{-3}\}$. We also have $\tau \in \{0.5, 0.7, 0.9\}$ and $\gamma \in \{0.1, 0.3, 0.5, 0.7, 0.9\}$.

**20 Newsgroups** For 20 Newsgroups, with the extracted features, we simply train a multilayer perceptron with two hidden layers of 300 neurons for 50 epochs. We use basically the same hyperparameters as for MNIST except that the learning rate $\alpha$ is selected from $\{5 \cdot 10^{-3}, 10^{-3}, 5 \cdot 10^{-4}\}$.

# D Additional Experiments

## D.1 Why Does PUbN\N Outperform nnPU ?

Our method, specifically designed for PUbN learning, naturally outperforms other baseline methods in this problem. Nonetheless, Table 1 equally shows that the proposed method when applied to PU learning, achieves significantly better performance than the state-of-the-art nnPU algorithm. Here we numerically investigate the reason behind this phenomenon.

Besides nnPU and PUbN\N, we compare with unbiased PU (uPU) learning (2). Both uPU and nnPU are learned with the sigmoid loss, learning rate $10^{-3}$ for MNIST, initial learning rate $10^{-4}$ for CIFAR-10, and learning rate $10^{-4}$ for 20 Newsgroups. This is because uPU learning is unstable with

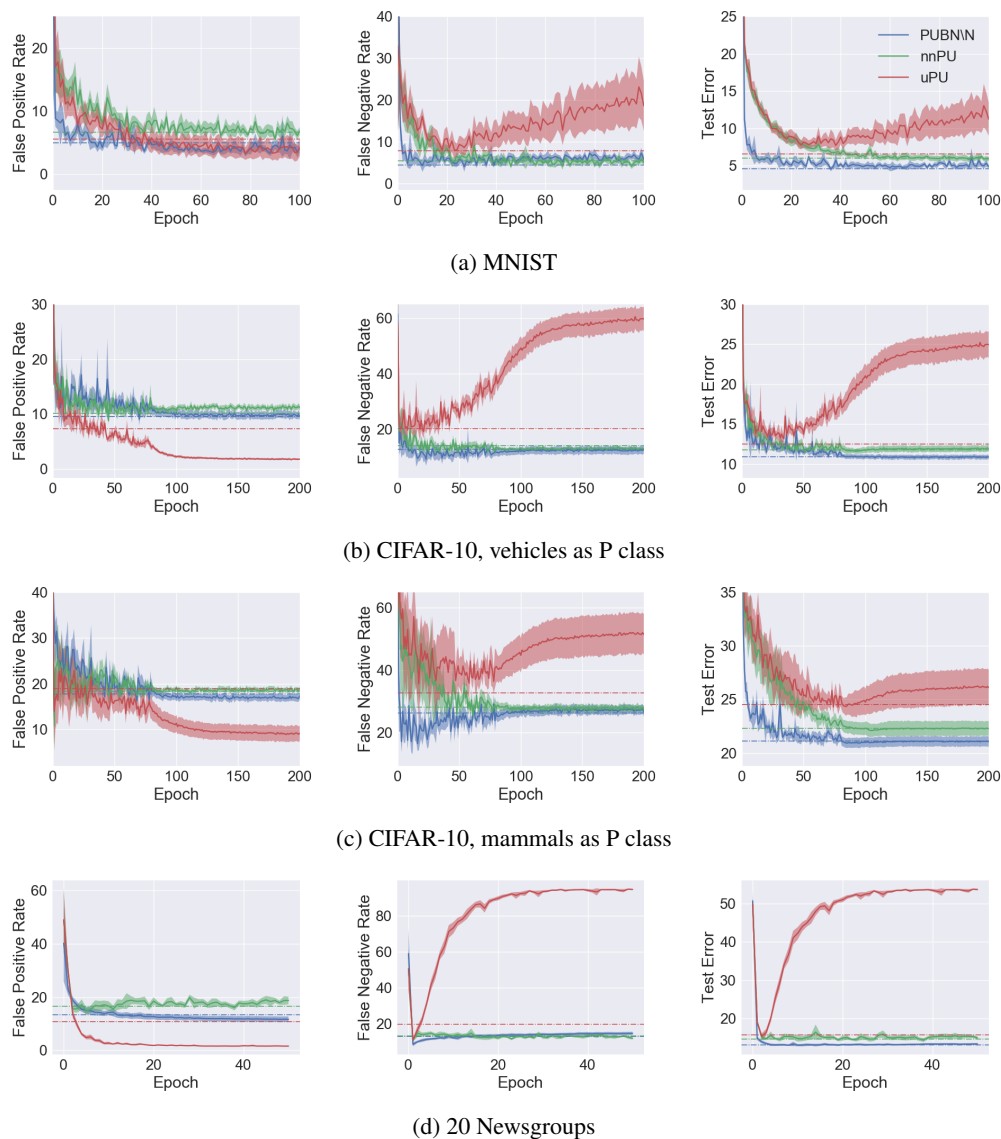

Figure 2: Comparison of uPU, nnPU and PUbN\N over the four PU learning tasks. For each task, means and standard deviations are computed based on the same 10 random samplings. Dashed lines indicate the corresponding values of the final classifiers (recall that at the end we select the model with the lowest validation loss out of all epochs).

the logistic loss. The other parts of the experiments remain unchanged. On the test sets we compute the false positive rates, false negative rates and misclassification errors for the three methods and plot them in Figure 2. We first notice that PUbN\N still outperforms nnPU trained with the sigmoid loss. In fact, the final performance of the nnPU classifier does not change much when we replace the logistic loss with the sigmoid loss.

In Kiryo et al. (2017), the authors observed that uPU overfits training data with the risk going to negative. In other words, a large portion of U samples are classified to the N class. This is confirmed in our experiments by an increase of false negative rate and decrease of false positive rate. nnPU remedies the problem by introducing the non-negative risk estimator (3). While the non-negative correction successfully prevents false negative rate from going up, it also causes more N samples to be classified as P compared to uPU. However, since the gain in terms of false negative rate is enormous, at the end nnPU achieves a lower misclassification error. By further identifying possible N samples after nnPU learning, we expect that our algorithm can yield lower false positive rate than

nnPU without misclassifying too many P samples as N as in the case of uPU. Figure 2 suggests that this is effectively the case. In particular, we observe that on MNIST, our method achieves the same false positive rate than uPU whereas its false negative rate is comparable to nnPU.

## D.2 Influence of $\eta$ and $\rho$

In the proposed algorithm we introduce $\eta$ to control how $\bar{R}_{s=-1}(g)$ is approximated from data and assume that $\rho = p(y = -1, s = +1)$ is given. Here we conduct experiments to see how our method is affected by these two factors. To assess the influence of $\eta$, from Table 1 we pick four learning tasks and we choose $\tau$ from $\{0.5, 0.7, 0.9, 2\}$ while all the other hyperparameters are fixed. Similarly to simulate the case where $\rho$ is misspecified, we replace it by $\rho' \in \{0.8\rho, \rho, 1.2\rho\}$ in our learning method and run experiments with all hyperparameters being fixed to a certain value. However, we still use the true $\rho$ to compute $\eta$ from $\tau$ to ensure that we always use the same number of U samples in the second step of the algorithm independent of the choice of $\rho'$.

The results are reported in Table 2 and Table 3. We can see that the performance of the algorithm is sensitive to the choice of $\tau$. With larger value of $\tau$, more U data are treated as N data in PUbN learning, and consequently it often leads to higher false negative rate and lower false positive rate. The trade-off between these two measures is a classic problem in binary classification. In particular, when $\tau = 2$, a lot more U samples are involved in the computation of the PUbN risk (7), but this does not allow the classifier to achieve a better performance. We also observe that there is a positive correlation between the misclassification rate and the validation loss, which confirms that the optimal value of $\eta$ can be chosen without need of unbiased N data.

Table 3 shows that in general slight misspecification of $\rho$ does not cause obvious degradation of the classification performance. In fact, misspecification of $\rho$ mainly affect the weights of each sample when we compute $\hat{R}_{\text{PUbN},\eta,\hat{\sigma}}$ (due to the direct presence of $\rho$ in (7) and influence on estimating $\sigma$). However, as long as the variation of these weights remain in a reasonable range, the learning algorithm should yield classifiers with similar performances.

## D.3 Estimating $\sigma$ from Separate Data

Theorem 2 suggests that $\hat{\sigma}$ should be independent from the data used to compute $\hat{R}_{\text{PUbN},\eta,\hat{\sigma}}$. Therefore, here we investigate the performance of our algorithm when $\hat{\sigma}$ and $g$ are optimized using different sets of data. We sample two training sets and two validation sets in such a way that they are all disjoint. The size of a single training set and a single validation set is as indicated in Appendix C.2, except for 20 Newsgroups we reduce the number of examples in a single set by half. We then use different pairs of training and validation sets to learn $\hat{\sigma}$ and $g$. For 20 Newsgroups we also conduct standard experiments where $\hat{\sigma}$ and $g$ are learned on the same data, whereas for MNIST and CIFAR-10 we resort to Table 1.

The results are presented in Table 4. Estimating $\sigma$ from separate data does not seem to benefit much the final classification performance, despite the fact that it requires collecting twice more samples. In fact, $\hat{\bar{R}}_{s=-1,\eta,\hat{\sigma}}^-(g)$ is a good approximation of $\bar{R}_{s=-1,\eta,\hat{\sigma}}^-(g)$ as long as the function $\hat{\sigma}$ is smooth enough and does not possess abrupt changes between data points. With the use of non-negative correction, validation data and L2 regularization, the resulting $\hat{\sigma}$ does not overfit training data so this should always be the case. As a consequence, even if $\hat{\sigma}$ and $g$ are learned on the same data, we are still able to achieve small generalization error with sufficient number of samples.

## D.4 Alternative Definition of nnPNU

In subsection 2.3, we define the nnPNU algorithm by forcing the estimator of the whole N partial risk to be positive. However, notice that the term $\gamma(1 - \pi)\hat{R}_{\text{N}}^-(g)$ is always positive and the chances are that including it simply makes non-negative correction weaker and is thus harmful to the final classification performance. Therefore, here we consider an alternative definition of nnPNU where we only force the term $(1 - \gamma)(\hat{R}_{\text{U}}^-(g) - \pi\hat{R}_{\text{P}}^-(g))$ to be positive. We plug the resulting algorithm in the experiments of subsection 4.2 and summarize the results in Table 5 in which we denote the alternative version of nnPNU by nnPU+PN since it uses the same non-negative correction as nnPU. The table indicates that neither of the two definitions of nnPNU consistently outperforms the other.

Table 2: Results on four different PUbN learning tasks when we vary the value of $\tau$ (and accordingly, $\eta$). Reported are means of false positive rates (FPR), false negative rates (FNR), misclassification rates (Error), and validation losses (VLoss) over 10 trials.

| Dataset | P | biased N | $\tau$ | FPR | FNR | Error | VLoss |
|---|---|---|---|---|---|---|---|
| MNIST | 2, 4, 6, 8, 10 | 1, 3, 5 | 0.5 | 4.79 | 4.32 | 4.56 | 10.11 |
| | | | 0.7 | 3.32 | 4.81 | 4.05 | **9.15** |
| | | | 0.9 | 3.29 | 4.40 | **3.83** | 9.30 |
| | | | 2 | 3.38 | 5.32 | 4.33 | 10.68 |
| CIFAR-10 | Airplane, automobile, ship, truck | Horse > deer = frog > others | 0.5 | 8.31 | 12.35 | **9.92** | **12.50** |
| | | | 0.7 | 8.23 | 13.15 | 10.20 | 12.62 |
| | | | 0.9 | 7.54 | 14.68 | 10.40 | 13.08 |
| | | | 2 | 6.23 | 20.29 | 11.85 | 13.64 |
| CIFAR-10 | Cat, deer, dog, horse | Bird, frog | 0.5 | 14.45 | 27.57 | 19.70 | 22.08 |
| | | | 0.7 | 13.20 | 27.27 | **18.83** | **20.72** |
| | | | 0.9 | 13.00 | 32.61 | 20.84 | 23.78 |
| | | | 2 | 11.67 | 31.49 | 19.60 | 22.52 |
| 20 Newsgroups | alt., comp., misc., rec. | soc. > talk. > sci. | 0.5 | 11.28 | 12.90 | **12.18** | **16.04** |
| | | | 0.7 | 11.40 | 13.58 | 12.62 | 16.64 |
| | | | 0.9 | 10.09 | 16.70 | 13.79 | 16.90 |
| | | | 2 | 10.34 | 20.55 | 16.06 | 20.99 |

Table 3: Mean and standard deviation of misclassification rates over 10 trials on different PUbN learning tasks when we replace $\rho$ by $\rho' \in \{0.8\rho, \rho, 1.2\rho\}$. Underlines indicate significant degradation of performance according to the 5% t-test.

| Dataset | P | biased N | $\rho'/\rho$ | | |
|---|---|---|---|---|---|
| | | | 0.8 | 1 | 1.2 |
| MNIST | 2, 4, 6, 8, 10 | 1, 3, 5 | $4.10 \pm 0.39$ | $4.05 \pm 0.27$ | $4.14 \pm 0.45$ |
| | | 9 > 5 > others | $3.85 \pm 0.55$ | $3.91 \pm 0.66$ | $3.94 \pm 0.54$ |
| CIFAR-10 | Airplane, automobile, ship, truck | Cat, dog, horse | $10.23 \pm 0.59$ | $9.71 \pm 0.51$ | $\underline{10.32 \pm 0.57}$ |
| | | Horse > deer = frog > others | $10.18 \pm 0.40$ | $9.92 \pm 0.42$ | $10.05 \pm 0.59$ |
| CIFAR-10 | Cat, deer, dog, horse | Bird, frog | $18.94 \pm 0.50$ | $18.83 \pm 0.71$ | $19.06 \pm 0.80$ |
| | | Car, truck | $20.39 \pm 1.24$ | $20.19 \pm 1.06$ | $19.92 \pm 0.89$ |
| 20 Newsgroups | alt., comp., misc., rec. | sci. | $13.49 \pm 0.61$ | $13.10 \pm 0.90$ | $13.31 \pm 1.05$ |
| | | talk. | $12.64 \pm 0.69$ | $12.61 \pm 0.75$ | $\underline{13.77 \pm 0.85}$ |
| | | soc. > talk. > sci. | $\underline{12.90 \pm 0.79}$ | $12.18 \pm 0.59$ | $\underline{12.74 \pm 0.35}$ |

It also ensures that there is always a clear superiority of our proposed PUbN algorithm compared to nnPNU despite its possible variant that is considered here.

Table 4: Mean and standard deviation of misclassification rates over 10 trials on different PUbN learning tasks with $\hat{\sigma}$ and $g$ trained using either the same or different sets of data.

| Dataset | P | biased N | Data for $\hat{\sigma}$ and $g$ | |
|---|---|---|---|---|
| | | | Same | Different |
| MNIST | 2, 4, 6, 8, 10 | 1, 3, 5 | $4.05 \pm 0.27$ | $3.71 \pm 0.45$ |
| | | 9 > 5 > others | $3.91 \pm 0.66$ | $4.06 \pm 0.36$ |
| CIFAR-10 | Airplane, automobile, ship, truck | Cat, dog, horse | $9.71 \pm 0.51$ | $10.00 \pm 0.51$ |
| | | Horse > deer = frog > others | $9.92 \pm 0.42$ | $9.66 \pm 0.46$ |
| CIFAR-10 | Cat, deer, dog, horse | Bird, frog | $18.83 \pm 0.71$ | $18.52 \pm 0.70$ |
| | | Car, truck | $20.19 \pm 1.06$ | $19.98 \pm 0.93$ |
| 20 Newsgroups | alt., comp., misc., rec. | sci. | $15.61 \pm 1.50$ | $16.60 \pm 2.38$ |
| | | talk. | $17.14 \pm 1.87$ | $15.80 \pm 0.95$ |
| | | soc. > talk. > sci. | $15.93 \pm 1.88$ | $15.80 \pm 1.91$ |

Table 5: Mean and standard deviation of misclassification rates over 10 trials on different PUbN learning tasks for the two possible definitions of the nnPNU algorithm.

| Dataset | P | biased N | nnPNU | nnPU + PN |
|---|---|---|---|---|
| MNIST | 2, 4, 6, 8, 10 | 1, 3, 5 | $5.33 \pm 0.97$ | $5.68 \pm 0.78$ |
| | | 9 > 5 > others | $4.60 \pm 0.65$ | $5.10 \pm 1.54$ |
| CIFAR-10 | Airplane, automobile, ship, truck | Cat, dog, horse | $10.25 \pm 0.38$ | $10.87 \pm 0.62$ |
| | | Horse > deer = frog > others | $9.98 \pm 0.53$ | $10.77 \pm 0.65$ |
| CIFAR-10 | Cat, deer, dog, horse | Bird, frog | $22.00 \pm 0.53$ | $21.41 \pm 1.01$ |
| | | Car, truck | $22.00 \pm 0.74$ | $21.80 \pm 0.74$ |
| 20 Newsgroups | alt., comp., misc., rec. | sci. | $14.69 \pm 0.46$ | $14.50 \pm 1.32$ |
| | | talk. | $14.38 \pm 0.74$ | $14.71 \pm 1.01$ |
| | | soc. > talk. > sci. | $14.41 \pm 0.70$ | $13.66 \pm 0.72$ |

