# OpenReview forum: "Classification from Positive, Unlabeled and Biased Negative Data"
_ICLR.cc/2019/Conference_

### Official Review · AnonReviewer1 · 2018-11-02
**I'm not convinced by the main assumption, it still needs more work to get published.**

**Rating:** 5
**Confidence:** 3

**Review:**

This paper has proposed a new algorithm for semi-supervised learning, which incorporate biased negative data into the existing PU learning framework.

The paper was written in clarity and easy to follow overall. However, the original motivation for having biased negative data are not explained very clear. The relation to dataset shift was very interesting, but it’s unclear what’s the exact connection between the proposed algorithm and the dataset shift. Maybe the authors can elaborate a little more on their point here in the future revision.

The paper has made some assumption about the relation between the latent random variable and the label in section 2.4. In the experiment, data sets are generated following the exact assumption. That’s not surprising to see that the proposed algorithm that fits the assumption will perform better than the previous methods without this assumption. In practice, there’s no way to really verify this assumption. Thus, it’s more interesting to see how the algorithm performs under the more generic semi-supervised learning setting, with unbiased, or biased negatives that don’t really fit the exact assumption in this paper.

Moreover, I’d like to see more intuition on why adding biased negative data will further improve upon nnPNU. The author provided some explanation in section 4.3, which seems just observations on the FPR and FNR, rather than the fundamental explanation for the advantage of this algorithm.

Choice of baseline methods is also limited. The original paper [1] for PNU has included a bunch of benchmark algorithms for semi-supervised learning. The authors should also include more benchmark algorithms for comparison, e.g. those listed in Section 5.2 in [1].

[1] Sakai, Tomoya, et al. "Semi-supervised classification based on classification from positive and unlabeled data." arXiv preprint arXiv:1605.06955 (2016).

---

> ### Author Response · Authors · 2018-11-12
> **Thank you for your insightful comments, we will address your concerns as quickly as possible**
>
> Dear Reviewer,
>
> Thank you very much for your insightful comments. Instead of answering all the questions at once, we will address the issues that you raise point by point, and also revise the paper accordingly.

---

> ### Author Response · Authors · 2018-11-12
> **1. Response to [Why adding biased negative data will further improve upon nnP(N)U.]**
>
> We suppose that you are asking why adding bN data improves upon "nnPU".
>
> This should be quite intuitive, because data explicitly labeled as N, even biased, should always carry some information and if we add them in the learning process it would certainly be beneficial, and that is also what motivated us to start studying this problem.
>
> Surely if this idea can be illustrated through figurer it will be clearer how bN data really helps learning a better classifier. We have therefore revised our paper to add corresponding content in Section 4.3 while the original Section 4.3 is now moved to appendix. We plot the representations learned by PUbN and nnPU classifiers and show that as expected, PUbN classifier can better separate P and bN data.
>
> - For the question "why does our method performs better than nnPNU"
> We think that is because the nnPNU algorithm does not take into account the fact that our labeled N data are biased.
>
> - The original section 4.3
> It studies the behavior of our algorithm when no bN data are available, so we do not think that it is related to you question. It is now moved to appendix because it is less related to the main problem studied by our paper.

---

> ### Author Response · Authors · 2018-11-12
> **2. Response to [Choice of baseline methods is also limited]**
>
> Semi-supervised baseline methods are not suitable in our case.
>
> In fact, as we have already highlighted in our abstract, the PUbN classification problem that we study is quite different from the semi-supervised learning setting because the labeled N distribution may not cover the whole N distribution. Take our MNIST experiment for example, in the first learning task only 1, 3 and 5 are labeled as N samples while the full N distribution also include 7 and 9.
>
> As a result, most of the semi-supervised learning algorithms cannot be directly applied to our problem. Ex: if we use entropy regularization for the above MNIST learning task with traditional PN risk, the accuracy is only around 80%; in fact, the regularization term cannot help at all if some subpopulation of data never appear in the labeled set.
>
> PNU learning is an exception because it relies partially on PU learning so it still somehow works in our problem. On the other hand, it is straightforward to combined different regularization-based semi-supervised learning algorithms with PUbN learning as it suffices to add the corresponding regularization term.

---

> ### Author Response · Authors · 2018-11-14
> **3. Response to [I'm not convinced by the main assumption]**
>
> We presume by the assumption in section 2.4 you are talking about the relation p(s=+1|x,y=+1)=1.
>
> This is in fact just a problem of notation. In other words, it is supposed that the marginal distribution p is a mixture of three component distributions q_1, q_2, q_3:
>
> p = a*q_1+b*q_2+c*q_3 with a, b, c >= 0 and a+b+c=1.
>
> Here q_1 is the P distribution, q_2 is the biased N distribution and (b*q_2+c*q_3)/(b+c) is the N distribution.
> Each time when x is drawn, we set the value of y and s according to the following rule
> - If x comes from q_1, y=s=+1,
> - If x comes from q_2, y=-1, s=+1,
> - If x comes from q_3, y=s=-1.
> (Here we would just like to give an intuition so we do not formulate the things mathematically.)
>
> We see the only assumption that is made is that the full N distribution is a mixture of the bN distribution and an unknown distribution q_3 (this is similar to the "selection condition" in sample selection bias). We believe that this is satisfied in many read-world scenarios because the bN samples are often collected/identified from a larger N data pool, which is the case in all the examples that are mentioned in our paper.
>
> In the very general case, bN and N distributions can differ arbitrarily and it is true that we cannot verify whether the above assumption is satisfied or not because we do not have access to samples that are sampled from the unbiased N distribution. Our algorithm is not guaranteed to work when the mixture assumption is violated. In fact, ρ cannot even be defined. One possible scenario is when some bN data belong to a latent category that does not appear in the true N distribution. Below are some preliminary experimental results:
>
> MNIST:
> [Positive [0, 2, 4, 6]; Negative [1, 3, 5, 7]; biased Negative [5, 7, 8, 9] (uniformly)]
> ------------------------------------
> nnPU       | 3.97 ± 0.66
> PUbN\N  | 3.26 ± 0.67
> PUbN      | 3.39 ± 0.77
> ------------------------------------
>
> CIFAR-10:
> [Positive [airplane, automobile, ship, truck]; Negative [bird, cat, deer, dog]; biased Negative [deer, dog, frog, horse] (uniformly)]
> ------------------------------------
> nnPU       | 12.63 ± 0.76
> PUbN\N  | 11.17 ± 0.39
> PUbN      | 10.51 ± 0.70
> ------------------------------------
>
> Reported are classification errors in percentage. Since ρ is ill-defined we arbitrarily choose ρ=0.25 to run our method. We see that in these two preliminary experiments the classifier learned with bN data works at least as well as that learned without bN data. This is because the U data are always exploited in our algorithm as long as ρ is not too large and η is not too small.
>
> We would however like to caution against applying our method directly without any modification if one knows that the [mixture assumption/selection condition] will be violated with great probability (though this should be rare). As already mentioned above, our algorithm is not guaranteed to converge to optimal solution and risks to perform worse than simple PU learning in this case.
>
> - Unbiased N data
>
> Our algorithm is designed particularly to deal with the bias of N data with help of the presence of U data. The U data are only used to correct the fact that our N data are biased, and no further assumption about the optimal model or the underlying distribution is made. Therefore, when the N data are unbiased, it falls back to classic supervised learning where U data are totally ignored. Ideas that allow to exploit U data to improve the classifier in the classic semi-supervised setting (N data being unbiased) can also be brought back to the PUbN case by simply replacing the PN risk by the PUbN risk.

---

> ### Author Response · Authors · 2018-11-14
> **4. Response to [The original motivation for having biased negative data are not explained very clear]**
>
> This part is already addressed in the introduction of the paper. In many PU learning problems, it is possible to collect some biased N data. However the presence of these non-representative N data are often ignored and we must resort to pure PU learning because few algorithms are able to leverage them in the learning process. Our proposed method then successfully incorporates these data into PU learning and improves the classification performance.
>
> One motivational situation is when the N population is formed by many subpopulations. It is normal that different subpopulations have different probabilities to get labeled as N. In [1], the authors mentioned the social media text classification problem where users are asked to manually labeled some relevant posts and irrelevant posts to a certain topic. Due to the inherent diversity of all the potentially irrelevant posts, the labeled N training posts cover only a small number of irrelevant topics and are therefore "biased".
>
> ----
> [1] G. Fei and B. Liu. Social media text classification under negative covariate shift. In Proceedings
> of the 2015 Conference on Empirical Methods in Natural Language Processing, pp. 2347–2356, 2015.

---

> ### Author Response · Authors · 2018-11-14
> **5. Response to [Relation to dataset shift]**
>
> Effectively we have mentioned that our problem setup can be viewed as a special case of dataset shift but we did not establish any connection between our method and any other algorithms dealing with the dataset shift problem.
>
> The reweighting technique is a popular solution to many related problems like covariate shift, imbalanced data or label noise. As a matter of fact, if we consider the case where "P and N distributions are disjoint, suppose that p(s=+1|x)>0 almost surely and set η=0", we recover the classic covariate shift reweighting scheme where no U data intervene in the second step of the algorithm when we estimate the classification risk. These conditions are however very restricted and in general our problem should not be compared with that of covariate shift.
>
> Pseudo-labeling is a well-known semi-supervised learning method and can also be applied to domain adaptation. The second step of our algorithm has some similarity with classic pseudo-labeling methods. In particular, we use a different base classifier (not the one trained with P and N data) to carefully assign pseudo labels to unlabeled samples and beyond this we also assign weights to each sample.
>
> The literature on dataset shift and related topics is so extensive that we cannot review all of the them here, but we believe the above two are the most related to our approach. Notice that as far as we know, no existing method designed for dataset shift is well-suited to solve the PUbN classification problem that we study in this paper.

---

### Official Review · AnonReviewer3 · 2018-11-04
**This paper studied classification problem, with Positive, Unlabeled and biased Negative labeled data.**

**Rating:** 6
**Confidence:** 3

**Review:**

This paper studied classification problem, with Positive, Unlabeled and biased Negative labeled data. The paper presents a two-step method, where the first-step is instance weighting and the second-step is standard binary classification. The paper shows theoretical proofs on the error estimation. Experiments on several well-known data sets are conducted and compared.

The good things of the paper are clear.

1.	Technical sound with statistical foundation
2.	Theoretical foundation
3.	Problem is general
4.	Paper is general well written.

Some weak points as well
1.	Application value is not so big, as there is no real application problem and the experiments are based on simulation.
2.	Although the studied problem is reasonable, the setup is a bit too general and need rather strict condition to have a good method.

---

> ### Author Response · Authors · 2018-11-14
> **Thank you for your feedback**
>
> Thank you very much for your feedback and approving that our paper is well written.
>
> To address the two weak points:
>
> We agree that better methods can be developed given stricter conditions or if the problem becomes more specific, but this is a trade-off that one always needs to face: a general setup with a good but not perfect algorithm or an ad hoc algorithm that works really well but only for rather restricted situations.
>
> Furthermore, we think there are also problems for which our method can be readily applied, namely the ones that are mentioned in our paper and in the replies to other reviewers.
> At the same time, the problem that we formulate and the approach that we consider can also inspire people who want to pursue in similar directions to design algorithms under stricter conditions or for more specific problems.

---

### Official Review · AnonReviewer2 · 2018-11-07
**Paper correct and carefully written but results rather straightforward**

**Rating:** 5
**Confidence:** 5

**Review:**

The authors rst present standard binary (positive negative or PN) classica-
tion, followed by positive unlabeled (PU) classication, that they motivate with
examples, such as one-class remote sensing classication. The new setting that
they introduce and study is called positive unlabeled biaised negative (PUbN
classication) and adds a biaised negative sample to PU learning. They give
motivating examples and compare this setting to the existing literature. A con-
vincing case is made regarding the dierence between the PUbN problem and
the known problems of semi-supervised learning and dataset shift.
They start by recalling the notations and nature of standard binary classi-
cation, PU classication and the nnPU (non-negative PU) strategy, as in the
previous PU learning papers. Then, they present the semi-supervised setting
under the name PNU learning, which simply studies the minimization of a con-
vex combination of the PN risk and the PU risk. As in PU learning, a correction
exists to avoid considering the estimate of the negative risk to be negative, re-
ferred to as nnPNU.
Finally, the authors introduce PUbN learning as the problem in which we
only have access to negatives that follow the law p(x\mid y = -1; s = +1), where s
is a latent variable that formalizes the bias.
As in PU learning, the authors derive an unbiased estimator of the risk that
involves only distributions for which data is available. However, they need
to reweight the P and bN distribution by the unknown posterior probability
sigma(x) = p(s = +1\mid x) of s. Considering s as the label, the problem of learning
a probabilistic classier separating the elements for which s = +1 and s = 􀀀1
can be seen as a PU learning problem, which gives an estimator ^ of sigma,
and makes the method practical.
They derive estimation error bounds, that depend on the mean squared
difference between sigma and sigma^ and a term of order n^-1/2 where n is the cardinal
of the smallest sample. They considered the function ^ as a xed function in
their bounds, which implies that the bounds are only true if some of the data is
kept for the estimation of sigma^. Finally, they present a variant of their algorithm
for PU learning, named PUbNnN where unlabeled instances are not all given
the same weight, but weighted according to sigma hat. The experiments use neural networks with stochastic optimization, on the classic datasets MNIST, CIFAR-10 and 20 Newsgroup. They report better per-
formance using their technique on all datasets. The authors documented their
experiences thoroughly in the appendix. However, I did not find information
about the nature of the estimator of the posterior probability sigma^, which is im-
portant for reproducibility. Furthermore, in appendix B, choosing sigma^ = 0 will
minimize the criterion . Finally, they proceed to justify the dominance of
the variant of their method over usual nnPU learning.

---

> ### Author Response · Authors · 2018-11-12
> **Thank you very much for your review, code to reproduce our results will be publicly available**
>
> Dear Reviewer2,
>
> Thank you very much for taking your time to review our paper. Your concise summary of our paper shall be very helpful to anybody who is interested in our work. Below we address the two concerns raised at the end of the review.
>
> ** 1. [Nature of the estimator of the posterior probability σ̂] **
>
> - As for the learning algorithm, as mentioned in your comment, we use s as label and trained a probabilistic classifier to separate s=+1 and s=-1 by leveraging PU learning algorithms. See section 3.1, Estimating σ:
> [In other words, here we regard X_P and X_bN as P and X_U as U, and attempt to solve a PU learning problem by applying nnPU.]
>
> - As for the model, we use always the same model for σ̂ and the main classifier. See section 4.1:
> [For simplicity, in an experiment, σ̂ and g always use the same model and are trained for the same number of epochs.]
>
> - By the way, we will put our code on github after the reviewing process to ensure that all the experiments in our paper can be easily reproduced by anybody who is interested.
>
> ** 2. [In appendix B, choosing σ̂=0 will minimize the criterion] **
>
> - This is not true, the criterion \hat{J} is minimized when σ̂~σ. The criterion is the empirical approximation of E_{x~p(x)}[|σ̂(x)-σ(x)|^2] plus some constant that is independent of σ̂. Due to the presence of this constant term, which is negative, \hat{J} can be negative and is not minimized when σ̂=0 in which case we have \hat{J}=0.
>
> Finally, we would like to thank you again for your detailed feedback and great summary of our paper.

---

### Meta-Review · Area_Chair1 · 2018-12-12
**Limited practical value**

**Confidence:** 4
**Recommendation:** Reject

**Metareview:**

The paper proposes an algorithm for semi-supervised learning, which incorporate biased negative data into the existing PU learning framework.

The reviewers and AC commonly note the critical limitation of practical value of the paper and results are rather straightforward.

AC decided the paper might not be ready to publish as other contributions are not enough to compensate the issue.